# √🌍 MATHNET: A GLOBAL MULTIMODAL BENCHMARK FOR MATHEMATICAL REASONING AND RETRIEVAL

**Shaden Alshammari**[1*]    **Kevin Wen**[1*]    **Abrar Zainal**[3*]    **Mark Hamilton**[1]

**Navid Safaei**[4]    **Sultan Albarakati**[2]    **William T. Freeman**[1†]    **Antonio Torralba**[1†]

[1] MIT    [2] KAUST    [3] HUMAIN    [4] Individual Researcher    [*†] Equal Contribution

√🌍 Website: `mathnet.mit.edu`    🤗 `shadena/mathnet`    ○ `shadealsha/mathnet`

Figure 1: **Overview of MATHNET.** MATHNET contains 30K+ Olympiad-level problems across 47 countries, 17 languages, and 143 competitions over 40 years with expert-authored solutions. We evaluate several leading models on problem solving and math-aware retrieval.

## ABSTRACT

Mathematical problem solving remains a challenging test of reasoning for large language and multimodal models, yet existing benchmarks are limited in size, language coverage, and task diversity. We introduce MATHNET, a high-quality, large-scale, multimodal, and multilingual dataset of Olympiad-level math problems together with a benchmark for evaluating mathematical reasoning in generative models and mathematical retrieval in embedding-based systems. MATHNET spans 47 countries, 17 languages, and two decades of competitions, comprising **30,676 expert-authored problems with solutions** across diverse domains. In addition to the core dataset, we construct a retrieval benchmark consisting of mathematically equivalent and structurally similar problem pairs curated by human experts.

MATHNET supports three tasks: (i) *Problem Solving*, (ii) , *Math-Aware Retrieval*, and (iii) *Retrieval-Augmented Problem Solving*. Experimental results show that even state-of-the-art reasoning models (78.4% for `Gemini-3.1-Pro` and 69.3% for `GPT-5`) remain challenged, while embedding models struggle to retrieve equivalent problems. We further show that RAG performance is highly sensitive to retrieval quality; for example, `DeepSeek-V3.2-Speciale` achieves gains of up to 12%, obtaining the highest scores on the benchmark. MATHNET provides the largest high-quality Olympiad dataset together with the first benchmark for evaluating mathematical problem retrieval, and we publicly release both the dataset and benchmark at `mathnet.mit.edu`.

# 1 INTRODUCTION

Recent large language models (LLMs) and large multimodal models (LMMs) have made rapid improvements on mathematical reasoning benchmarks, from grade-school problems to competition mathematics (Cobbe et al., 2021; Hendrycks et al., 2021b; Achiam et al., 2023). Recently, public reports claimed unprecedented gold-medal–level performance at the International Mathematical Olympiad (IMO) by advanced multiple models (Luong et al., 2025; Shao et al., 2025). Moreover, there have been multiple incidents of AI systems reportedly solving open mathematical problems (Nie et al., 2025; Feldman & Karbasi, 2025).

Despite these advances, the lack of open, high quality, and diverse benchmarks constrains research progress. Existing Olympiad-level datasets are typically drawn from community platforms such as AoPS and cover only a handful of competitions in the U.S and China (see Table 1). To address this gap, we present MATHNET: a large-scale, multimodal, and multilingual collection of Olympiad-level mathematics problems sourced from 47 countries across four decades. The full dataset collection, MATHNET, contains 30K+ problems with *high-quality official solutions* written by experts across a wide range of mathematical domains. Its scale, diversity, and expert quality provide an unprecedented foundation for exploring mathematical generalization and analogical reasoning.

We use MATHNET to study two main capabilities: *Problem Solving*, or the ability to solve mathematical problems, and *Math-Aware Retrieval*, or the ability to recognize and retrieve mathematically equivalent or related problems. In particular, unlike existing semantic retrieval (Izacard et al., 2021; Khattab & Zaharia, 2020; Formal et al., 2021), our problem retrieval task must be aware of symbolic structure, invariances, and transformations. For example, the problem of solving $x^2 + y^2 = 1$ is equivalent to $\sqrt{a^2 + b^2} = 1$, and is also equivalent to the set of 2D vectors with unit norm $|u|^2 = 1$. Crucially, these are not equivalent to solving $x + y = 1$. Current retrieval models fail to make this distinction: due to superficial lexical overlap (Das et al., 2025), they often rank a problem containing $x + y = 1$ as closer to $x^2 + y^2 = 1$ than to the truly equivalent formulations. Despite the foundational importance of math-aware retrieval, we note this task remains largely unexplored in recent literature.

These challenges arise even in expert workflows such as the annual IMO exam selection process. During shortlist construction, problems may sometimes resemble problems that already exist in books, problem collections, or online sources, illustrating how difficult it can be to recognize mathematical equivalence across different notations, formats, and languages. Similar issues arise in mathematical research. For example, a mathematician studying bounds on gaps between consecutive primes may search for phrases such as "upper bounds on prime gaps" rather than a specific formula like $p_{n+1} - p_n \leq C(\log p_n)^2$ (where $p_n$ is the $n$-th prime and $C$ is a constant). However, existing retrieval systems are often sensitive to superficial features such as variable naming or textual phrasing, making it difficult to connect mathematically equivalent statements expressed in different forms.

To make progress on these challenges, we introduce MATHNET, a collection of mathematics problems of unprecedented size supporting model analysis across three tasks: (i) *Problem Solving*, (ii) *Math-Aware Retrieval*, and (iii) *Retrieval-Augmented Problem Solving*. Our contributions are:

1. **Main Corpus.** `MathNet-Solve`, a 30K-problem collection of Olympiad-level math with aligned LaTeX and natural-language statements, expert solutions, and metadata spanning 47 countries, 17 languages, and 65+ mathematical domains.

2. **Datasets for Retrieval.** `MathNet-Retrieve`, a dataset for *Math-Aware Retrieval* comprised of 40K additional synthetic problems derived from 10K anchor problems, each paired with 1 equivalent positive and 3 hard negatives. `MathNet-RAG`, a dataset for *Retrieval-Augmented Problem Solving* built from 70 IMO-level expert-curated structurally similar problems.

3. **Benchmark Evaluation.** Benchmarking across 27 state-of-the-art models on three primary benchmarks: *Problem Solving* accuracy on `MathNet-Solve`, *Math-Aware Retrieval* using Recall@k on `MathNet-Retrieve`, and *Retrieval-Augmented Problem Solving* accuracy on `MathNet-RAG`, using both automatic grading and human expert grading.

4. **Analysis: Solving vs. Retrieving.** We demonstrate embedding model performance in *Math-Aware Retrieval* lags behind LLM and LMM performance in *Problem Solving*. Moreover, for *Retrieval-Augmented Problem Solving*, retrieval-augmented generation (RAG) improves reasoning only when retrievers surface structure-aligned, mathematically relevant neighbors.

## 2 RELATED WORK

Mathematical problem solving has long been a core benchmark for evaluating AI reasoning capabilities. Early efforts focused on text-based arithmetic problems, while recent research has expanded to competition-level reasoning, theorem proving, and multimodal problem-solving. Existing datasets can be broadly categorized into text-only benchmarks, multimodal benchmarks, and aggregates.

**Text-Only Mathematical Benchmarks.** Several datasets evaluate LLMs' mathematical reasoning using text-only problems. Cobbe et al. (2021) introduced **GSM8K**, grade-school level problems for elementary arithmetic reasoning. Hendrycks et al. (2021b) proposed **MATH**, which consists of problems spanning high school to competitive mathematics. Gao et al. (2024b) presented **Omni-MATH**, with 4,428 Olympiad-level problems. He et al. (2024) and Wang et al. (2024) further extend coverage with bilingual and competition-level datasets, though most are limited in scale, language diversity, or structured similarity annotations.

**Multimodal Mathematical Benchmarks.** Multimodal benchmarks integrate visual information with textual descriptions, primarily for geometry or diagram-based reasoning. Datasets such as **MATH-Vision** (Wang et al., 2024) and **MathVista** (Lu et al., 2024) incorporate broad visual contexts, including charts and diagrams. Despite this added modality, these datasets remain comparatively easy and do not capture the full difficulty of Olympiad-level problem solving. **Large-Scale Aggregates.** Large datasets aggregate problems from multiple sources such as NuminaMath (Li et al., 2024b) and (Albalak et al., 2025). Although valuable for large-scale training and evaluation, these datasets typically lack curated multimodal content, multi-lingual coverage, and fine-grained annotations.

**Math-Aware Retrieval.** There has been work on formula-aware indexing (Zanibbi et al., 2025; Das et al., 2025), but such systems predate LLMs and typically operate at the formula level, missing broader conceptual and structural similarities expressed in natural language. Meanwhile, modern IR excels at semantic paraphrase but is often *blind* to symbolic equivalence and cross-modal cues.

**Limitations of Prior Work and Motivation for** MATHNET. Despite these advances, current benchmarks exhibit three main limitations: (i) limited expert solutions, (ii) lack of visual multilingual content, especially for high-difficulty problems, and (iii) limited analysis of mathematical problem retrieval. MATHNET addresses these gaps by offering a large-scale, multilingual, multimodal dataset of Olympiad-level problems. It includes expert-validated problem pairs and a fine-grained taxonomy of mathematical similarity, enabling rigorous study of analogical problem solving in generative models, retrieval quality in embedding-based systems, and retrieval-augmented reasoning across languages and modalities.

| Benchmark | Size | Languages | Evaluation Type | M | Source | Difficulty |
|---|---|---|---|---|---|---|
| GSM8k Cobbe et al. (2021) | 8,500 | EN | Numeric Answer | × | Crowdsourced problems | 🟩 Grade School |
| MATH Hendrycks et al. (2021b) | 12,500 | EN | Numeric Answer | × | Competitions / textbooks | 🟨 High School |
| MATH-Vision Wang et al. (2024) | 3,040 | EN | Expression / Proof | ✓ | Math Competitions | 🟧 High School |
| CMMLU Li et al. (2024a) | 11,528 | ZH | MCQ | × | Chinese exam materials | 🟨 High School / College |
| MMLU Hendrycks et al. (2021a) | 15,908 | EN | MCQ | × | College / professional exams | 🟧 College-Level |
| C-Eval Huang et al. (2023) | 13,948 | ZH | MCQ | × | Chinese college exams | 🟧 College Entrance |
| MMMU Yue et al. (2024) | 11,500 | EN | MCQ / Expression | ✓ | Multimodal academic exams | 🟧 College-Level |
| AGIEval Zhong et al. (2024) | 3,300 | EN & ZH | MCQ / Expression | × | College entrance exams | 🟧 College Entrance |
| JEEBench Arora et al. (2023) | 515 | EN | MCQ / Numeric Answer | × | Indian JEE Advanced | 🟧 JEE Advanced Exam |
| OlympiadBench He et al. (2024) | 6,142 | EN & ZH | Proof / Expression | ✓ | Official Websites | 🟥 Olympiad Level |
| OlympicArena Huang et al. (2024) | 3,233 | EN & ZH | Proof / Process | ✓ | Official Websites | 🟥 Olympiad Level |
| Omni-Math Gao et al. (2024b) | 4,428 | EN | Proof / Process | × | AoPS Forum / Contest Pages | 🟥 Olympiad Level |
| IneqMath Sheng et al. (2025) | 1,552 | EN | Proof / Analytical Tools | × | Curated Inequalities Problems | 🟥 Olympiad Level |
| OlymMATH Sun et al. (2025) | 200 | EN & ZH | Numeric Answer | × | AoPS Forum/Official Websites | 🟥 Olympiad Level |
| LiveAoPS Mahdavi et al. (2025) | - | EN | Numeric / Expression | × | AoPS Forum (rolling snapshot) | 🟥 Olympiad Level |
| MathArena Balunović et al. (2025) | 162 | EN | Final Answer / Proof | ✓ | Newly released competitions | 🟥 Olympiad Level |
| IMOBench Luong et al. (2025) | 460 | EN | Numeric / Proof | × | IMO & national archives | 🟥 Olympiad Level |
| **MathNet** (ours) | **30,767** | EN, ZH, ES RU, FR, RO and 11 other | Expression / Proof | ✓ | Official Country Booklets/ International and National Contests | 🟥 Olympiad Level |

Table 1: Comparison of mathematical reasoning benchmarks across different sizes, languages, evaluation types, and difficulty levels. We include both unimodal and multimodal datasets, spanning grade-school to Olympiad-level mathematics. Our Proposed MATHNET expands coverage to 17 languages and focuses on proof- and process-based evaluation with national contest problems.

## 3 DATASET AND BENCHMARK DESIGN

MATHNET consists of Olympiad-level problems together with benchmarks in three tasks that evaluates mathematical reasoning in generative models, mathematical retrieval in embedding-based systems, and analogical reasoning with retrieval augmented generation.

A key feature of MATHNET is its fine-grained taxonomy of mathematical similarity, which enables systematic analysis of both solver and retriever performance across varying levels of structural and semantic overlap. To complement the dataset, we define a novel retrieval task that measures whether embedding-based systems can identify related problems based on deeper structural relationships rather than surface-level features. We further provide baselines and evaluations for both generative reasoning and retrieval, demonstrating the utility of the benchmark built on top of the dataset.

### 3.1 DATASETS AND TASKS

MATHNET consists of three datasets: `MathNet-Solve`, `MathNet-Retrieve`, and `MathNet-RAG`. The associated tasks are *Problem Solving*, *Math-Aware Retrieval*, and *Retrieval-Augmented Problem Solving*. Benchmarks combine a task, a dataset, and an evaluation criterion.

1. `MathNet-Solve`: 30,676 expert-written Olympiad problems with solutions, spanning 47 countries, 17 languages, and 143 competitions. Divided into 3 splits: `MathNet-Solve-train` (23776 samples), and `-test` (6400 samples). Serves as the source collection from which retrieval datasets are built. Supports *Problem Solving* via generated solutions graded against reference solutions. Figure 2 summarizes main statistics of the corpus.

2. `MathNet-Retrieve`: an evaluation dataset for *Math-Aware Retrieval* built from 10,000 anchor problems from `MathNet-Solve`. For each anchor, we construct 1 equivalent positive and 3 hard negatives, yielding 40,000 synthetic problems in total. Used to evaluate embedding models on retrieving mathematically equivalent problems.

3. `MathNet-RAG`: an evaluation dataset of 35 anchor problems and 35 expert-paired real problems all drawn entirely from `MathNet-Solve`; it is used for *Retrieval-Augmented Problem Solving* under both retrieved and oracle contexts.

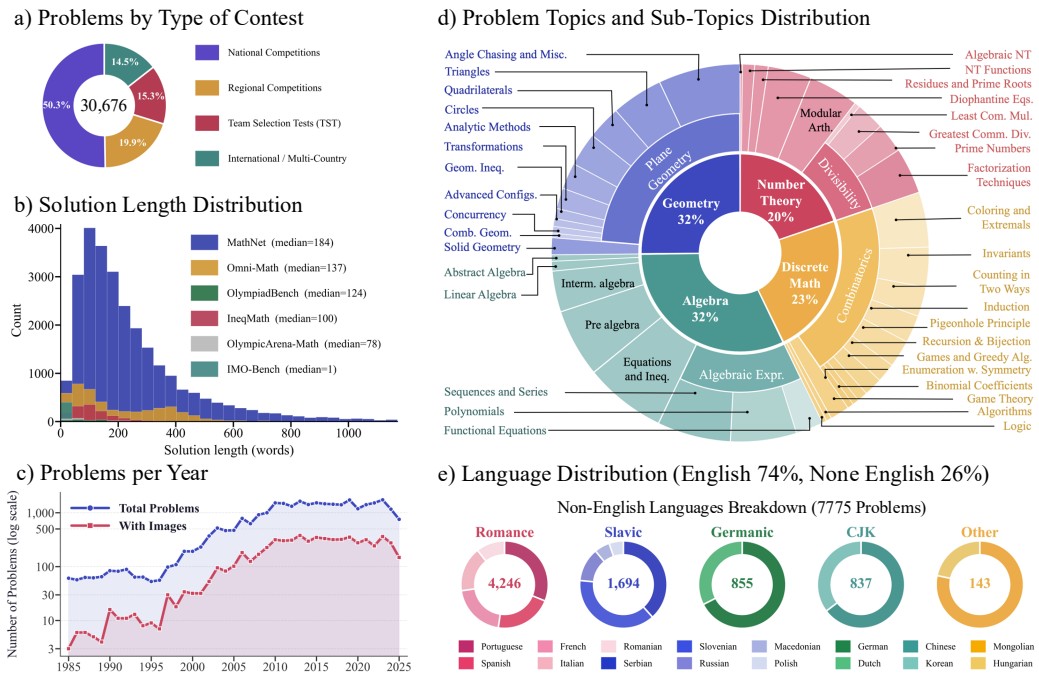

Figure 2: **Overview of `MathNet-Solve`.** The dataset spans national, regional, TST, and international competitions, with varying solution lengths. It has grown since the early 2000s and includes textual and diagram-based problems with broad multilingual and topical coverage.

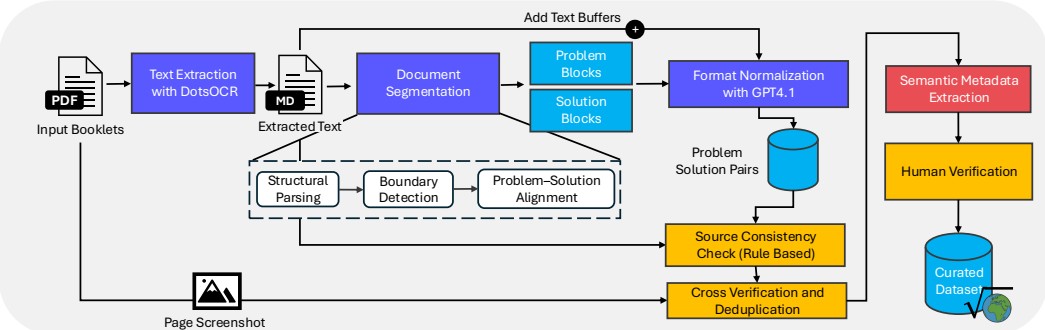

Figure 3: Overview of MATHNET problem–solution extraction pipeline. The curation pipeline consists of three stages: (1) document ingestion and problem segmentation, (2) problem and solution extraction with format normalization, and (3) multi-stage extraction verification.

## 3.2 DATA COLLECTION, EXTRACTION AND ANNOTATION

**Data sources.** Each year, participating IMO countries contribute original problems for use in their national contests and team selection examinations. To construct the dataset underlying the benchmark, we collect official problem booklets from 47 countries spanning 1985–2025, comprising 1595 PDF volumes and more than 25,000 pages in total. Unlike prior math benchmarks that often rely on community-sourced platforms such as AoPS, MathNet is built exclusively from officially published national materials. All included problems and solutions are authored and disseminated by national teams themselves, ensuring expert-level quality, consistency in style, and immunity from noisy or informal annotations. For more details, refer to section A.

**Problems Extraction.** We first convert all contest booklets into a Markdown format using the multilingual document parsing framework `dots-ocr` (Li et al., 2025; Zheng et al., 2026). This step establishes a uniform input format for downstream processing. The underlying source material spans a wide range of formats including both digital typeset documents and scanned copies, with some documents spanning several languages. We leverage the multilingual recognition and layout analysis capabilities of `dots-ocr` to robustly handle this variation and ensure consistent and faithful text extraction across diverse document types.

**Problem-Solution Matching and Annotation** In general, it is a difficult problem to extract aligned problem-solution pairs from large corpora of heterogeneous mathematical documents. In our dataset, some documents present problems and solutions in separate sections, while others interleave them. Numbering schemes and naming conventions vary not only across countries but often within a single document. These inconsistencies render traditional parsing techniques (e.g., regex-based heuristics) brittle and difficult to scale. To address this, we designed a novel LLM-based pipeline for problem–solution alignment (illustrated in Figure 3). Our approach operates in three stages:

**Stage 1: Document Ingestion and Problem Segmentation.** To process each contest booklet, we first extract section-level segments, then pass these pages to `Gemini-2.5-Flash` to identify problem and solution corresponding segments by outputting only their line numbers. We also record the problem authors, hints and remarks (if present), as well as the and source file and page number to maintain provenance information for each problem.

**Stage 2: Problem and Solution Extraction.** Given the line segments from Stage 1, we extract the text within these segments, along with additional buffer text before and after them in case any content is missing. We then use the prompt `GPT-4.1` to extract the corresponding problem and solution in a LaTeX-friendly format. This strategy addresses the issue of problems and solutions appearing in different sections of the booklet, which may exceed the extraction model's context window.

**Stage 3: Extraction Verification.** We validate the extracted problem–solution pairs through a three-stage verification process of (1) a rule-based analytical checker, (2) `GPT-4.1`, and (3) human annotators. First, we compute text similarity between the extracted content and the original OCR output to ensure that LLM editing is restricted to formatting adjustments and introduces no hallucinated content. Second, we prompt `GPT-4.1` to compare screenshots of the source pages with the extracted problem–solution pairs, acting as a judge (see Appendix F) to detect OCR errors, incorrect figure associations, or mismatches between problems and solutions, and determine whether the solution is complete. Finally, we manually review cases with low confidence scores. A problem–solution pair is retained only if all three verification mechanisms reach unanimous agreement.

| Mode | Problem A | Problem B |
|---|---|---|
| **Invariance** | | |
| Syntactic Equivalence | Find $f : \mathbb{R} \to \mathbb{R}$ such that $f(x^2 - y^2) = (x - y)(f(x) + f(y))$. | Find $g : \mathbb{R} \to \mathbb{R}$ such that $(g(a) + g(b))(a - b) = g(a^2 - b^2)$. |
| Reformulation | Let $a_i > 0$. Prove $\sum_{i=1}^n \frac{a_i}{a_i^2 + a_{i+1}a_{i+2}} \leq \sum_{i=1}^n \frac{1}{a_i + a_{i+1}}$ | Let $a_i > 0$. Prove $\sum_{i=1}^n \frac{a_i^2}{a_i^2 + a_{i+1}a_{i+2}} \geq \frac{1}{2}$ |
| Transformational | Find all $x \in \mathbb{R}$ such that $4^x + 6^x = 9^x$. | Find all $x \in \mathbb{R}$ such that $(2/3)^x + (3/2)^x = 5/2$. |
| **Structural Resonance** | | |
| Generalization | For $k \geq 1$, prove that $k$ divides $\binom{n}{k}$ for all $n \geq k$. | Show that $\binom{n}{m} \equiv \prod \binom{n_i}{m_i} \pmod{p}$, where $n = \sum n_i p^i$, $m = \sum m_i p^i$. |
| Common Lemma | If $ab + 1 | a^2 + b^2$, show that $\frac{a^2 + b^2}{ab + 1}$ is a perfect square. | If $a^2 + b^2 + c^2 = k(ab + bc + ca)$, show that $k \in \{1, 2, 3\}$. |
| Structural Reduction | Prove that $4^n + 2^n + 1$ is never a prime number. | Prove that $2^{2n} + 2^n + 1$ is divisible by 3 for all $n$. |
| **Affinity** (Thematic) | Show that the largest prime factor of $\binom{2n}{n}$ is greater than $n^{2/3}$. | For every $n > 1$, there is a prime $p$ such that $n < p < 2n$. |

Table 2: **Taxonomy of mathematical similarity with Olympiad-style examples.** Invariance captures strict equivalence under reformulation, Structural Resonance reflects shared lemmas or reductions, and Affinity denotes looser thematic clustering.

## 3.3 RETRIEVAL IN MATHEMATICS

### 3.3.1 UNDERSTANDING DIFFERENT TYPES OF MATHEMATICAL SIMILARITY

Mathematical progress often depends on recognizing when different problems share common structure. These structural commonalities can take several forms, from strict equivalence to looser thematic connections. We distinguish three modes of similarity: *Invariance*, *Resonance*, and *Affinity*. For examples see Table 2.

**Invariance** refers to strict equivalence under transformation. Two problems are invariant when they differ only in representation but share the same underlying structure. Examples include syntactic renaming, algebraic reformulation, geometric re-characterization, or cross-domain isomorphism.

**Resonance** refers to partial similarity. Problems are not identical, but they can be addressed using the same idea, proof strategy, or structural analogy. Resonance highlights opportunities to transfer tools or insights across contexts.

**Affinity** refers to a broad sense of relatedness without structural equivalence. Problems may belong to the same conceptual or disciplinary area (e.g., number theory, geometry) even if they do not share a method or solution strategy. Affinity provides a way to group problems by theme, context, or historical development.

### 3.3.2 BUILDING MATHNET-RETRIEVE

We constructed this dataset to support *Math-Aware Retrieval*, whcih aims to distinguish between surface-level lexical overlap and deep mathematical equivalence. `MathNet-Retrieve` is built from 10,000 anchor problems drawn from `MathNet-Solve`. For each anchor, we use `GPT-5` (prompt details in the Appendix) to generate exactly 1 equivalent positive and 3 hard negatives, yielding 40,000 synthetically derived problems in total.

**Equivalent Positives.** For each anchor problem, we generate one mathematically equivalent variant via variable renaming (e.g., $x \to a$), algebraic manipulation, and paraphrasing using `GPT-5` (prompt details in the Appendix). For example, the functional equation $f(x) + f(y) = f(x + y)$ can be rewritten as an algebraically equivalent variant such as $g(a) - g(a + b) = -g(b)$.

**Hard Negatives.** For each anchor, we also generate three adversarial hard negatives that preserve much of the surface form while changing the underlying mathematics (e.g., $f(x^2) + f(y) = f(x - y)$). These near-miss distractors make it difficult for models to succeed by relying only on lexical similarity.

### 3.3.3 BUILDING MATHNET-RAG

We also construct `MathNet-RAG` as a non-synthetic evaluation dataset of expert-paired real Olympiad problems, supporting *Retrieval-Augmented Problem Solving*. The dataset is organized as 35 anchors paired with 35 expert-curated pairs, 70 problems total drawn from real Olympiad competitions. Each pair lies in the *Structural Resonance* category of our taxonomy and exhibits similarities such as *Generalization*, *Common Lemma*, or *Structural Reduction*. These pairs capture a human expert notion of similarity that goes substantially beyond simple algebraic transformation.

> **Example Conceptual Problem Pair from `MathNet-RAG`**
>
> **Problem A.** [Chinese TST 2014] Show that there are no 2-tuples $(x, y)$ of positive integers satisfying
>
> $$(x+1)(x+2)\cdots(x+2014) \;=\; (y+1)(y+2)\cdots(y+4028).$$
>
> **Problem B.** [Russia 2009] Alireza multiplied a billion consecutive natural numbers, and Matin multiplied two million consecutive natural numbers. Prove that these two got different results.

## 4 EXPERIMENTS

### 4.1 EXPERIMENTAL SETUP

We evaluate 27 models on MATHNET using three benchmarks: *Problem Solving* accuracy on `MathNet-Solve`, *Math-Aware Retrieval* using Recall@k on `MathNet-Retrieve`, and *Retrieval-Augmented Problem Solving* accuracy on `MathNet-RAG`.

On `MathNet-Solve`, we evaluate two types of models: (i) **LLMs and LMMs**, including `gpt-4o`, `Llama-4-Maverick-17B-128E-Instruct-FP8`, and `Grok-3`. For models that accept images, we provide both the text and image as input; otherwise, we supply a text-only description of the image. (ii) **LLMs and LMMs with deliberate reasoning**, including `gemini-3.1-pro-preview`, `gemini-3-flash-preview`, `gemini-2.5-flash`, `claude-opus-4.6`, `gpt-5`, `gpt-5-mini`, `gpt-5-nano`, and `DeepSeek-R1`.

On `MathNet-Retrieve`, we evaluate retrieval performance using embeddings derived from a diverse set of state-of-the-art models, including `all-mpnet-base-v2`, `multi-qa-mpnet-base-dot-v1`, `cohere-embed-v4.0`, `qwen3-embedding-4B`, `gemini-embedding-001`, `text-embedding-ada-002`, `text-embedding-3-small`, and `text-embedding-3-large`. We compute similarities between problem statements using cosine similarity over the embedding representations.

On `MathNet-RAG`, we limit the evaluations to seven state-of-the-art open-source and proprietary models, as this benchmark requires human grading. The models are `gemini-3-pro-preview`, `gpt-5`, `claude-opus-4.5`, `DeepSeek-V3.2-Speciale`, `oLMO-3-Think`, `Grok-4.1-Fast`, and `Phi-4-reasoning-plus`. For each model we evaluate three inference settings: (i) Zero Shot, where the model receives only the target problem; (ii) Embed-RAG, where we retrieve one related problem using embeddings from `gemini-embedding-001` and provide that retrieved problem together with its official solution; and (iii) Expert-RAG, where we instead provide the expert-paired related problem from `MathNet-RAG` together with its official solution.

### 4.2 EVALUATION PROTOCOL

***Problem Solving*** on `MathNet-Solve`. Similar to the protocol introduced in IMO-Bench (Luong et al., 2025), we adopt a score-based model grading procedure using `GPT-5`. For each problem, the judge model is provided with the problem statement, the reference solution, and the model-generated solution, and is asked to assess whether the output is consistent with the correct answer using a numeric score from 0–7. We binarize the score by marking outputs with score $\geq 6$ as correct (fully correct or containing only minor errors) and scores $< 6$ as incorrect. This allows us to distinguish between models that arrive at the correct final answer by coincidence versus those that demonstrate consistent reasoning ability. We also report performance by subject domain (algebra, geometry, combinatorics, number theory), enabling a fine-grained analysis of model strengths and weaknesses.

***Math-Aware Retrieval*** **on `MathNet-Retrieve`.** The primary evaluation metric for our retrieval task is **Recall@k**, which measures whether any of the top-$k$ retrieved problems correspond to a "correct" match from our equivalent versions of each problem. We report Recall@1 and Recall@5. To better understand embedding behavior, we further analyze cosine similarity distributions between equivalent problem pairs, unrelated pairs, and near misses (hard negatives), highlighting cases where models struggle to separate fine-grained distinctions.

***Retrieval-Augmented Problem Solving* on `MathNet-RAG`.** To assess the impact of retrieval on downstream problem solving, we evaluate three settings. In Zero Shot, the model receives only the target problem. In Embed-RAG, we retrieve one related problem using `gemini-embedding-001`, then provide the retrieved problem and its official solution as additional context. In Expert-RAG, we replace the retrieved example with the expert-paired related problem from `MathNet-RAG`, again together with its official solution. Comparing Zero Shot, Embed-RAG, and Expert-RAG isolates the effect of retrieval quality: the gap from Zero Shot to Embed-RAG measures the value of retrieved context from an embedding retriever, while the gap from Embed-RAG to Expert-RAG measures how much performance is still limited by retrieval errors.

## 4.3 MAIN RESULTS

### 4.3.1 PROBLEM SOLVING ON MATHNET-SOLVE

Table 3 reports *Problem Solving* accuracy on `MathNet-Solve` across four core domains: Algebra, Number Theory, Geometry, and Discrete Mathematics. The strongest overall model is `gemini-3.1-pro`, which achieves 76.3% overall accuracy, followed by `gemini-2.5-pro` at 71.9%, and `gpt-5` and `gemini-3-flash-preview`, both at 68.1%. Across models, Algebra is consistently the easiest domain, with the top systems reaching 82.9%, while Geometry and Discrete Mathematics remain the most challenging: even `gpt-5`, despite strong Algebra (79.4%) and Number Theory (73.0%) performance, drops to 56.3% on Geometry and 64.1% on Discrete Mathematics.

A clear performance stratification emerges. Frontier reasoning models—especially the Gemini 3.1/2.5 and GPT-5 families—substantially outperform earlier or smaller models. Mid-tier systems such as `claude-opus-4-6`, `gpt-5-nano`, `DeepSeek-V3.2`, and `gemini-2.5-flash` achieve overall scores between 38.8% and 43.9%, while weaker baselines including `grok-3`, `DeepSeek-V3-0324`, `gpt-4.1`, `Llama-4-Maverick-17B-128E-Instruct-FP8`, `gpt-4o`, and `Ministral-3B` lag far behind. Notably, the gap between the top and bottom of the table is large: `gemini-3.1-pro-preview` outperforms `Ministral-3B` by 72.7 points overall. Overall, these results highlight that Olympiad-level mathematical reasoning remains challenging even for state-of-the-art systems, with the largest appearing in Geometry and Discrete Mathematics. For more detailed breakdowns by language and multimodality, see Appendix D.

| *Problem Solving* Accuracy (%, ↑) on `MathNet-Solve-Test` | | | | | | |
|---|---|---|---|---|---|---|
| | Algebra | Number Theory | Geometry | Discrete Math | **Macro Avg.** | **Micro Avg.** |
| **LLMs (text-only)** | | | | | | |
| `ministral-3B` | $6.4 \pm 1.0$ | $2.9 \pm 0.9$ | $4.3 \pm 0.8$ | $1.7 \pm 0.6$ | $4.4 \pm 0.5$ | $4.4 \pm 0.5$ |
| `DeepSeek-V3.2` | $51.6 \pm 2.0$ | $45.3 \pm 2.6$ | $32.2 \pm 1.8$ | $32.7 \pm 2.1$ | $40.1 \pm 1.2$ | $40.1 \pm 1.2$ |
| `grok-3` | $37.7 \pm 1.9$ | $33.0 \pm 2.3$ | $21.7 \pm 1.6$ | $24.2 \pm 1.9$ | $28.5 \pm 1.1$ | $28.5 \pm 1.1$ |
| **LMMs (text + images)** | | | | | | |
| `Llama-4-Maverick-17B*` | $22.5 \pm 1.7$ | $14.4 \pm 1.8$ | $10.7 \pm 1.2$ | $8.6 \pm 1.3$ | $14.7 \pm 0.9$ | $14.7 \pm 0.9$ |
| `gpt-4.1` | $29.4 \pm 1.8$ | $24.0 \pm 2.2$ | $15.7 \pm 1.4$ | $16.6 \pm 1.7$ | $21.4 \pm 1.0$ | $21.4 \pm 1.0$ |
| `gpt-4o` | $10.9 \pm 1.2$ | $7.0 \pm 1.3$ | $4.5 \pm 0.8$ | $4.2 \pm 0.9$ | $6.8 \pm 0.6$ | $6.8 \pm 0.6$ |
| **LLMs with reasoning** | | | | | | |
| `DeepSeek-R1` | $46.1 \pm 2.0$ | $39.5 \pm 2.5$ | $31.2 \pm 1.8$ | $27.3 \pm 2.0$ | $36.3 \pm 1.2$ | $36.3 \pm 1.2$ |
| **LMMs with reasoning** | | | | | | |
| `gemini-3.1-pro-preview` | $\mathbf{83.7 \pm 1.5}$ | $\mathbf{82.2 \pm 2.0}$ | $\mathbf{74.6 \pm 1.7}$ | $\mathbf{75.6 \pm 2.0}$ | $\mathbf{78.4 \pm 1.0}$ | $\mathbf{78.4 \pm 1.0}$ |
| `gemini-3-flash-preview` | $\underline{77.7 \pm 1.7}$ | $\underline{73.3 \pm 2.3}$ | $\underline{67.0 \pm 1.8}$ | $64.0 \pm 2.2$ | $\underline{70.4 \pm 1.1}$ | $\underline{70.4 \pm 1.1}$ |
| `gemini-2.5-flash` | $50.5 \pm 2.0$ | $42.6 \pm 2.5$ | $36.8 \pm 1.8$ | $31.0 \pm 2.1$ | $41.1 \pm 1.2$ | $41.1 \pm 1.2$ |
| `gpt-5` | $80.3 \pm 1.6$ | $73.6 \pm 2.3$ | $61.1 \pm 1.9$ | $65.3 \pm 2.2$ | $69.3 \pm 1.1$ | $69.3 \pm 1.1$ |
| `gpt-5-mini` | $67.6 \pm 1.8$ | $61.5 \pm 2.6$ | $50.3 \pm 2.0$ | $50.2 \pm 2.3$ | $57.0 \pm 1.2$ | $57.0 \pm 1.2$ |
| `gpt-5-nano` | $53.9 \pm 2.0$ | $49.6 \pm 2.6$ | $32.4 \pm 1.8$ | $34.6 \pm 2.1$ | $42.2 \pm 1.2$ | $42.2 \pm 1.2$ |
| `claude-opus-4.6` | $53.2 \pm 2.0$ | $44.6 \pm 2.5$ | $44.3 \pm 1.9$ | $36.4 \pm 2.2$ | $45.7 \pm 1.2$ | $45.7 \pm 1.2$ |

Table 3: ***Problem Solving* on `MathNet-Solve-Test` (6,400 problems).** Results are reported as accuracy (%) $\pm$ standard error across four domains; models are grouped by modality and reasoning capability, with **best results** in bold and second-best results underlined. **Takeaway:** LMMs with reasoning achieve the strongest performance overall, while Geometry and Discrete Mathematics remain the most challenging domains.

### 4.3.2 Math-Aware Retrieval on MathNet-Retrieve

As shown in Table 4, *Math-Aware Retrieval* on MathNet-Retrieve remains highly challenging at the top-1 level, with even the strongest models (Qwen3-embedding-4B and Gemini-embedding-001) achieving only ~5% Recall@1. Performance improves markedly at higher cutoffs, with Recall@10 exceeding 80% in several domains. Among all models, Gemini-embedding-001 provides the most consistent gains, delivering the highest Recall@5 and Recall@10 across domains and the strongest aggregate performance (68.88% and 83.79%, respectively). In contrast, legacy embedding models such as text-embedding-ada-002 and text-embedding-3-small perform substantially worse across all settings.

These results suggest that current general-purpose embedding models fail to capture the deep structural and symbolic relationships that define mathematical equivalence. A critical failure mode is that both LLMs and LMMs often rely on superficial textual overlap (e.g., matching on keywords such as "triangle" or "polynomial") rather than reasoning over the underlying mathematical concepts. The weak top-1 retrieval performance highlights that these models lack a robust internal representation of mathematical knowledge that would support analogical reasoning across problem variants. This gap underscores the need for embeddings explicitly trained to encode mathematical structure, rather than depending on incidental surface-level cues.

To further illustrate this issue, Figure 6 shows the distribution of cosine similarities between equivalent and non-equivalent problems. Surprisingly, non-equivalent pairs often exhibit higher similarity scores than equivalent ones. This counterintuitive trend highlights that embeddings frequently capture superficial lexical or symbolic overlap rather than true structural relationships, leading models to mis-rank distinct problems as closer than genuinely equivalent ones. This explains the weak Recall@1 performance observed in Table 4.

| Math-Aware Retrieval Recall@k (%, ↑) on MathNet-Retrieve | | | | | | | | | | |
|---|---|---|---|---|---|---|---|---|---|---|
| **Model** | **Algebra** | | **Number Theory** | | **Geometry** | | **Discrete Math** | | **All** | |
| | **R@1** | **R@5** | **R@1** | **R@5** | **R@1** | **R@5** | **R@1** | **R@5** | **R@1** | **R@5** |
| all-mpnet-base-v2 | 4.54 | 73.06 | 4.67 | 82.54 | 4.37 | 74.76 | 4.25 | 75.38 | 3.78 | 57.70 |
| multi-qa-mpnet-base-dot-v1 | 4.00 | 69.40 | 3.73 | 80.76 | 3.88 | 71.73 | 3.98 | 73.40 | 3.27 | 55.08 |
| cohere-embed-v4.0 | 2.73 | 59.85 | 2.67 | 68.85 | 2.35 | 59.87 | 2.78 | 63.40 | 2.24 | 44.81 |
| qwen3-embedding-4B | 5.24 | 78.74 | 4.62 | 86.43 | **5.60** | 79.05 | **5.96** | 81.50 | **4.96** | 64.95 |
| gemini-embedding-001 | **5.50** | **81.62** | **4.95** | **87.43** | 5.49 | **81.86** | 5.35 | **82.80** | 4.83 | **68.88** |
| text-embedding-ada-002 | 2.05 | 54.94 | 2.22 | 63.35 | 2.16 | 55.07 | 2.71 | 57.51 | 1.94 | 42.02 |
| text-embedding-3-small | 2.10 | 47.47 | 1.89 | 54.62 | 2.10 | 47.61 | 2.84 | 50.12 | 1.98 | 35.49 |
| text-embedding-3-large | 3.19 | 68.18 | 2.73 | 75.25 | 3.20 | 68.18 | 3.35 | 69.52 | 2.74 | 54.23 |

Table 4: *Math-Aware Retrieval* **on MathNet-Retrieve (10,000 anchor problems).** Results are reported as Recall@1 and Recall@5 across four domains and overall, with **best results** in bold and second-best results underlined. **Takeaway:** Across domains, Recall@1 remains low even for the strongest models, while gemini-embedding-001 and qwen3-embedding-4B achieve the best overall Recall@5, highlighting that current embedding models retrieve mathematically equivalent problems only reliably at larger retrieval depths.

### 4.3.3 Retrieval-Augmented Problem Solving on MathNet-RAG

As shown in Table 5, Expert-RAG is the strongest setting overall, but the gains are not uniform across models or grading protocols. Under human grading, DeepSeek-V3.2-Speciale reaches the best result in the table at 97.3% with Expert-RAG, and GPT-5 also improves substantially from 76.8% in Zero Shot to 86.6% in Expert-RAG. At the same time, the table also shows that stronger retrieved context does not guarantee gains for every model: for example, Gemini-3-Pro is best under human grading in Zero Shot and Embed-RAG, while Claude-4.5-Opus and oLMO-3-Think show smaller or mixed changes under human grading. Under average LLM grading, the same overall pattern holds: Expert-RAG is often competitive or best, but Embed-RAG is less reliable and can fall below Zero Shot, as seen for Grok-4.1-Fast, Gemini-3-Pro, and oLMO-3-Think. Taken together, the table suggests that retrieval helps most when the added example is truly structure-aligned. Embed-RAG can help, but its benefits are inconsistent because embedding-based retrieval sometimes returns near-miss problems that add noise rather than useful mathematical guidance.

| | | Retrieval-Augmented Problem Solving Accuracy (%, ↑) on MathNet-RAG | | | | | |
|---|---|---|---|---|---|---|---|
| **Model** | **RD** | **Human Grading** | | | **LLM Grading** | | |
| | **(2025)** | **Zero-shot** | **Embed-RAG** | **Expert-RAG** | **Zero-shot** | **Embed-RAG** | **Expert-RAG** |
| `DeepSeek-V3.2-Speciale` | 01 Dec | $84.8 \pm 6.1$ | $89.5 \pm 5.2$ | $\mathbf{97.3 \pm 2.7}$ | $82.2 \pm 6.5$ | $\mathbf{87.9 \pm 5.5}$ | $\mathbf{89.0 \pm 5.3}$ |
| `Claude-4.5-Opus` | 24 Nov | $46.8 \pm 8.4$ | $55.5 \pm 8.4$ | $52.4 \pm 8.4$ | $46.0 \pm 8.4$ | $50.3 \pm 8.5$ | $56.4 \pm 8.4$ |
| `oLMO-3-Think` | 20 Nov | $45.2 \pm 8.4$ | $54.6 \pm 8.4$ | $47.6 \pm 8.4$ | $49.5 \pm 8.5$ | $45.6 \pm 8.4$ | $51.1 \pm 8.4$ |
| `Grok-4.1-Fast` | 19 Nov | $75.4 \pm 7.3$ | $83.8 \pm 6.2$ | $83.2 \pm 6.3$ | $73.1 \pm 7.5$ | $67.7 \pm 7.9$ | $69.1 \pm 7.8$ |
| `Gemini-3-Pro` | 18 Nov | $\mathbf{89.1 \pm 5.3}$ | $\mathbf{92.9 \pm 4.3}$ | $\underline{87.5 \pm 5.6}$ | $73.2 \pm 7.5$ | $70.5 \pm 7.7$ | $76.4 \pm 7.2$ |
| `GPT-5` | 07 Aug | $76.8 \pm 7.1$ | $75.2 \pm 7.3$ | $86.6 \pm 5.8$ | $\mathbf{87.1 \pm 5.7}$ | $\underline{81.8 \pm 6.5}$ | $\underline{85.8 \pm 5.9}$ |
| `Phi-4-Reasoning Plus` | 30 Apr | $15.1 \pm 6.1$ | $14.3 \pm 5.9$ | $16.7 \pm 6.3$ | $24.1 \pm 7.2$ | $19.6 \pm 6.7$ | $30.0 \pm 7.7$ |

Table 5: **Retrieval-Augmented Problem Solving on `MathNet-RAG` (35 problems).** Results are reported as accuracy (%) ± standard error under human grading and average LLM grading across three inference settings: zero-shot, Embed-RAG, and Expert-RAG; **best results** are shown in bold and second-best results are underlined. **Takeaway:** Expert-RAG yields the most consistent gains, but improvements remain model-dependent and grading-dependent.

## 5 Discussion and Limitation

Results on MATHNET reveal a clear gap between strong generative performance on *Problem Solving* and weak performance on *Math-Aware Retrieval* for mathematical equivalence. While frontier LLMs and LMMs achieve impressive scores on answer-generation benchmarks, our retrieval results show that current embedding-based systems still struggle to capture mathematical structure. The limited gains from visual augmentation further suggest that multimodal integration for symbolic tasks remains underdeveloped.

The strong performance of the formula-aware baseline indicates that structured, non-textual representations are crucial for retrieval. Progress in true mathematical reasoning may require moving beyond next-token prediction toward architectures that explicitly integrate symbolic reasoning.

## 6 Conclusion

In this work, we introduced MATHNET to analyze current models in *Problem Solving*, *Math-Aware Retrieval*, and *Retrieval-Augmented Problem Solving*. By providing a corpus of 30,676 problems with a fine-grained taxonomy of equivalence, we enabled a rigorous study of mathematical generalization and analogical reasoning. To ensure reliability, we complemented automated extraction with systematic human validation: expert annotators reviewed problem similarity labels, and student evaluators assessed the alignment and completeness of extracted problem–solution pairs. These human contributions establish a strong ground-truth foundation, ensuring that MATHNET captures deep mathematical structure rather than superficial overlap.

Our comprehensive evaluations show that while frontier generative models can solve complex problems, current retrieval systems still struggle with a fundamental yet overlooked task: retrieving mathematically equivalent or structurally related problems from large corpora. This deficiency in retrieval highlights a key limitation in current learned representations of mathematical knowledge. We hope MATHNET will serve as a valuable resource for the community, paving the way for research into improved retrieval-augmented reasoning, symbolic AI, and ultimately, more capable and reliable problem-solving models.

## 7 Acknowledgments

Shaden acknowledges support from the Schwarzman College of Computing Fellowship. This work was also supported by the National Science Foundation under Cooperative Agreement PHY-2019786 (NSF AI Institute for Artificial Intelligence and Fundamental Interactions; http://iaifi.org/).

The authors thank the International Mathematical Olympiad (IMO) President, Gregor Dolinar, and the IMO Foundation Board Chair, Maria Losada, for their guidance and support, and for facilitating communication with IMO country team leaders to collect recent problems. We are also grateful to the graders who generously volunteered their time to evaluate solutions and provide feedback: Smbat Gogyan[1], Dominik Burek[2], Majid Almarhoumi[3], Alzubair Habibullah[4], Melih

Ucer[5], Hamza Alshaikhi[6], Khursav Yorov[7], Omar Habibullah[8], Mohammed Alghamdi[9], Yosuf Bakheet[10], Hikmatullo Ismatov[11], Asaad Saleh[12], Ali Alramadan[13], and Ibraheem Khan[14].

We further acknowledge the significant effort of Navid Safaei in collecting a substantial portion of this archive from past IMOs (since 2006), including manually scanning many of the documents. This project would not have been possible without his dedication and willingness to share these materials with the broader research and education community. The authors would also like to thank: Anne Harrington, Neha Hulkund, and Elena Sierra for their valuable discussions and Jay Sekora from MIT for helping setting up the website.

[1] Armenia Team Leader and IMO Silver 1998  [2] Poland Team Leader  [3] Saudi Deputy Leader and IMO Bronze 2016  [4] Saudi Deputy Leader and IMO Silver 2017 [5] IMO Gold 2010 / Top 4 Scores  [6] IMO Bronze 2021  [7] Math Ph.D. candidate and Olympiad Coach  [8] IMO Silver 2019  [9] IMO Bronze 2025  [10] IMO Bronze 2025  [11] Math PhD Candidate and Olympiad Coach  [12] IMO Bronze 2019  [13] IMO Bronze 2023  [14] IMO Bronze 2014

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

## APPENDIX

This appendix contains additional results, dataset examples, tables, figures, prompts, and implementation details used throughout the paper. We include these materials to support reproducibility and provide further analysis beyond the main text.

## A   OVERVIEW OF COMPETITIONS COVERED BY MATHNET

This section summarizes the competitions represented in `MathNet-Solve`. The dataset includes problems drawn from a wide range of mathematical olympiads, together with related training and team selection materials.

Table 6 lists all sources included in the dataset and the years covered. The first part of the table contains international and multi-country competitions (e.g., the Asia Pacific Mathematics Olympiad, the Balkan Mathematical Olympiad, and Baltic Way). The second part summarizes national sources grouped by country, including national mathematical olympiads, team selection tests for international competitions (e.g., IMO team selection tests), and related national contests and training materials.

| Source | Years | Competitions |
|---|---|---|
| **National competitions and team selection tests by country** | | |
| Argentina | 2003–2024 | Argentina Mathematical Olympiad (Argentine MO); Cono Sur Mathematical Olympiad; Rioplatense Mathematical Olympiad; Iberoamerican Mathematical Olympiad; May Mathematical Olympiad |
| Australia | 2010–2024 | Australian Mathematical Olympiad; Australian Intermediate Mathematics Olympiad (AIMO); Australian Mathematical Olympiad Committee Senior Contest (AMOC Senior Contest); APMO; EGMO; IMO; Mathematics Ashes; Mathematics Challenge for Young Australians (MCYA) |
| Austria | 2010–2024 | Austrian Mathematical Olympiad (Regional Round; Preliminary Round; Final Round; Beginners' Competition) |
| Belarus | 2010–2025 | Belarusian Mathematical Olympiad; IMO TST; Selection and Training Session |
| Brazil | 2006–2022 | Brazilian Mathematical Olympiad (OBM), Galois–Noether University Mathematics Competition |
| Bulgaria | 2003–2024 | Bulgarian Mathematical Olympiad; Bulgarian Autumn Tournament; Bulgarian Spring Tournament; Bulgarian Winter Tournament; BMO TST; IMO TST; JBMO TST |
| Canada | 1969–2025 | Canadian Mathematical Olympiad (CMO) |

| Source | Years | Competitions |
| --- | --- | --- |
| China | 2007–2025 | China Mathematical Olympiad (CMO); China Mathematical Competition; China TST; China Girls' Mathematical Olympiad (CGMO); China Western Mathematical Olympiad; China Southeastern Mathematical Olympiad; China Western Invitational Mathematical Competition |
| Croatia | 2011–2019 | Croatian Mathematical Olympiad; Croatian Junior Mathematical Olympiad; Croatia National Mathematical Competition (City Round, County Round, Final Round); MEMO TST; IMO TST |
| Czech Republic | 2000–2025 | Czech Mathematical Olympiad; Czech and Slovak Mathematical Olympiad; Czech–Polish–Slovak Mathematical Match; Czech–Slovak Match |
| Estonia | 2010–2025 | Estonian MO; Estonian Open Contests; IMO TST |
| France | 2011–2024 | French Mathematical Olympiad; EGMO TST; French Mathematical Olympiad Preparation |
| Germany | 2000–2022 | German Mathematical Olympiad; IMO TST |
| Greece | 2007–2024 | Hellenic Mathematical Olympiad (Archimedes); Hellenic Mathematical Olympiad; BMO; JBMO; Mediterranean Mathematical Competition; IMO TST |
| Hong Kong | 1997–2024 | Hong Kong (China) Mathematical Olympiad (HKMO); Preliminary Selection Contest; Pre-IMO Mock Exam; IMO TST |
| India | 2000–2024 | Indian National Mathematical Olympiad (INMO); Indian Regional Mathematical Olympiad (RMO); EGMO TST; IMO TST; IMO Training Camp Practice Test |
| Iran | 2010–2025 | Iranian Mathematical Olympiad, IMO TST |
| Ireland | 2007–2025 | Irish Mathematical Olympiad |
| Italy | 1998–2024 | Italian Mathematical Olympiad; Italian Mathematical Olympiad Project |
| Japan | 2006–2024 | Japanese Mathematical Olympiad (JMO); Japanese Junior Mathematical Olympiad (JJMO) |
| Kazakhstan | 2014–2021 | International Zhautykov Olympiad |
| Moldova | 2023 | Moldova Mathematical Olympiad |
| Mongolia | 2010–2025 | Mongolian Mathematical Olympiad; IMO TST; EGMO TST |
| Netherlands | 2006–2025 | Dutch Mathematical Olympiad; Dutch Junior Mathematical Olympiad; BxMO; BxMO/EGMO TST; IMO TST |
| New Zealand | 2019–2025 | New Zealand Mathematical Olympiad |
| Macedonia | 2008–2023 | Macedonian Mathematical Olympiad; Junior Macedonian Mathematical Olympiad; BMO; JBMO; Mediterranean Mathematical Olympiad; European Mathematical Cup; IMO TST |
| Philippines | 2018 | Philippine Mathematical Olympiad |
| Poland | 2023–2025 | Polish Mathematical Olympiad |
| Portugal | 2016–2017 | Olympic Revenge |
| Romania | 2004–2025 | Romanian Mathematical Olympiad; RMM; IMAR Mathematical Competition; Stars of Mathematics Competition; Danube Mathematical Competition; Clock-Tower School Competition; Selection Tests for BMO, JBMO, and IMO; Romanian Mathematical Olympiad Shortlist; Gazeta Matematică National Olympiad |
| Russia | 2009–2025 | Russian Mathematical Olympiad; All-Russian Mathematical Olympiad; Euler Olympiad; EGMO TST |
| Saudi Arabia | 2010–2025 | Saudi Arabian Mathematical Competitions; BMO TST; EGMO TST; Gulf Mathematical Olympiad TST (GMO TST); IMO TST; JBMO TST; Camp Tests; Preselection Test |

| Source | Years | Competitions |
|---|---|---|
| Serbia | 2007–2023 | Serbian Mathematical Olympiad; District Mathematics Competition; Municipal Mathematics Competition; IMO TST |
| Singapore | 2010–2025 | Singapore Mathematical Olympiad (SMO); Singapore Mathematical Olympiad Open (SMO Open); Singapore International Math Olympiad Challenge (SIMOC); IMO Training Camp |
| Slovenia | 2001–2023 | Slovenian Mathematical Olympiad; Slovenian Mathematical Olympiad for Technical Schools; IMO TST |
| South Africa | 2010–2024 | South African Mathematical Olympiad (SAMO); Rhodes University Camp; University of Stellenbosch Camp; South African Talent Search; South African Monthly Problem Sets; IMO TST |
| South Korea | 2006–2024 | Korean Mathematical Olympiad (KMO) |
| Soviet Union | 1961–1991 | All-Union Mathematical Olympiad; CIS Mathematical Olympiad |
| Spain | 2001–2023 | Spanish Mathematical Olympiad; Iberoamerican Mathematical Olympiad; Mediterranean Mathematical Olympiad; Barcelona Contest Preparation; BarcelonaTech Mathcontest; IMO |
| Switzerland | 1999–2023 | Swiss Mathematical Olympiad; IMO TST |
| Taiwan | 2011–2024 | Taiwan Mathematical Olympiad; APMO Taiwan Preliminary Round; Taiwan IMO Team Selection Training Camp; Taiwan Mathematical Olympiad Training Camp |
| Thailand | 2007–2017 | Thailand Mathematical Olympiad; Thailand Third Mathematical Olympiad (T3MO); Thailand Training Camp |
| Turkey | 2000–2024 | Turkish Mathematical Olympiad; Turkish Junior Mathematical Olympiad; EGMO TST; JBMO TST; IMO TST |
| Ukraine | 2005–2022 | Ukrainian Mathematical Olympiad; Ukrainian National MO |
| United Kingdom | 2006–2022 | British Mathematical Olympiad (BMO) |
| United States | 1985–2025 | American Mathematics Competitions 10 and 12 (AMC 10/12); American Invitational Mathematics Examination (AIME); United States of America Mathematical Olympiad (USAMO); United States of America Junior Mathematical Olympiad (USAJMO); IMO TST; Harvard–MIT Mathematics Tournament (HMMT); Bay Area Mathematical Olympiad (BAMO); Berkeley Math Circle; William Lowell Putnam Mathematical Competition |
| Vietnam | 2001–2024 | Vietnamese MO; Mock Test; Prep Test; IMO TST |
| **International and Multi-country Competitions** | | |
| APMO | 1989–2025 | Asia Pacific Mathematics Olympiad |
| BMO (SL) | 2004–2025 | Balkan Mathematical Olympiad & Shortlist |
| BW (SL) | 1990 | Baltic Way & Shortlist |
| BxMO | 2010–2025 | Benelux Mathematical Olympiad |
| CPS | 2005–2025 | Czech–Polish–Slovak Mathematical Match |
| EGMO | 2012–2025 | European Girls' Mathematical Olympiad |
| IMO (SL) | 2006–2024 | International Mathematical Olympiad & Shortlist |
| JBMO | 2003–2023 | Junior Balkan Mathematical Olympiad |
| MEMO | 2008–2024 | Middle European Mathematical Olympiad |
| NMC | 1987–2024 | Nordic Mathematical Contest |
| OIM | 1985–2019 | Ibero-American Mathematical Olympiad |
| RMM | 2010–2021 | Romanian Master of Mathematics |
| SRMC | 2002–2025 | Silk Road Mathematics Competition |
| IZhO | 2014–2021 | International Zhautykov Olympiad |

Table 6: Competitions and Countries included in MATHNET.

## B  TAXONOMY OF TOPICS COMMONLY USED IN MATH OLYMPIAD

We provide the curated taxonomy used for labeling domains, subjects, topics, and subtopics. These labels ground our analyses and enable consistent cross-competition comparisons.

| Sub-subtopic | Key Concepts |
| --- | --- |
| **Geometry** | |
| **Plane Geometry** | |
| Triangles | Centroid, incenter, circumcenter, orthocenter, ex-centers, Euler line, nine-point circle; geometric inequalities; trigonometry (metric relations) |
| Quadrilaterals | Cyclic, inscribed/circumscribed, Complete quadrangle, perpendicular diagonals |
| Circles | Angles, coaxal, tangents, radical axis, metric relations, Apollonius circle |
| Concurrency / Collinearity | Theorems of Ceva, Menelaus, Pappus, Desargues |
| Transformations | Translation, rotation, homothety, spiral similarity, inversion, the method of moving points |
| Advanced Configurations | Simson line, Miquel, Napoleon / Fermat / Brocard points, symmedians, polar triangles, harmonic/isogonal/isotomic conjugates, barycentric coordinates |
| Geometric Inequalities | Classical and advanced |
| Combinatorial Geometry | Helly, Sylvester, convex hulls, Pick theorem, Minkowski theorem, convex figures |
| Analytic / Coordinate Methods | Complex numbers, Cartesian coordinates, vectors, trigonometric relations |
| Miscellaneous | Angle/distance chasing, constructions, loci |
| **Solid Geometry** | |
| 3D Shapes | Polyhedra, prisms, pyramids, spheres, cylinders, cones |
| Volume | Cavalieri's principle, Formulae and problem-solving |
| Surface Area | Formulae and applications |
| Other 3D problems | Mixed problems, reducing the problem into a plane geometry problem |
| **Differential Geometry** | |
| Curvature | Gaussian, mean |
| Manifolds | Surfaces, parametric |
| Geodesics | Shortest paths, great circles |
| **Non-Euclidean Geometry** | |
| Spherical Geometry | Spherical triangles, angles, area |
| Hyperbolic Geometry | Lines, models, inequalities |
| | |
| **Algebra** | |
| **Prealgebra / Basic Algebra** | |
| Integers | Sets of integers, Divisibility, primes, the Greatest Common Divisor (GCD), the Least Common Multiplier (LCM) |
| Fractions | Operations, simplification, comparison |
| Decimals | Conversion, operations, rounding |
| Simple Equations | Linear equations, word problems |
| Other | Number properties, prime factorization, divisors |
| **Algebraic Expressions** | |

| Sub-subtopic | Key Concepts |
|---|---|
| Polynomials | Operations, factorization, Algebraic identities, symmetric functions, Vieta's formula, interpolation formulae, complex numbers, roots of unity, Chebyshev polynomials and other trigonometric polynomials, irreducibility of polynomials, Descartes rule of signs, rootso of polynomials, Intermediate Value Theorem (IVT) |
| Sequences / Series | Recurrences, Charachteristic equations, monotonocity, boundedness, periodicity, convergence and divergence, floors/ceilings, sums/products, telescoping sums, Abel summation |
| Functional Equations | Substitution, defining a new function, Cauchy's equations, Injectivity/surjectivity, Periodicity, application of Calculus and Mathematical Analysis, iterations |
| **Inequalities** | |
| Functional considerations | Linear/Quadratic solving techniques |
| Classical inequalities | Cauchy-Schwarz, QM-AM-GM-HM, Power Mean, Jensen's Inequality, smoothing, Muirhead, Chebyshev's inequality, majorization, combinatorial optimization |
| **Discrete Mathematics** | |
| **Graph Theory** | |
| Basic concepts | Vertices, edges, path, connected graphs, cycles, Hamiltonian cycle and path, trees |
| Matchings | Marriage Lemma, Tutte's theorem |
| Connectivity | Menger, max-flow min-cut |
| Extremal | Turán |
| Euler characteristic | $V - E + F$ |
| **Combinatorics** | |
| Enumeration | Symmetry, basic counting techniques, recursion, bijection, inclusion-exclusion, double counting |
| Probability | Expected values, probabilistice methods, partitions, generating functions |
| Binomial coefficients | Algebraic properties |
| Pigeonhole principle | Applications |
| Invariants / Monovariants | Problem-solving |
| Coloring / Extremal | Graph problems |
| Induction | Standard and smoothing |
| Games / Greedy | Strategies, combinatorial games |
| **Logic / Algorithms / Other** | |
| Logic | Propositional/predicate logic, truth tables |
| Algorithms | Sorting, searching, Dynamic Programming (DP), greedy |
| Other | Miscellaneous problems, strategy development problems, inter-deciplinary problems |

| **Number Theory** | |
|---|---|
| **Divisibility / Factorization** | |
| Primes | Properties, sieves, prime numbers tests |
| GCD | Euclidean algorithm; linear combinations; Bezout's identity |
| LCM | Computation; relation with GCD |
| Factorization | Trial, Fermat, Pollard |
| **Modular Arithmetic** | |

| Sub-subtopic | Key Concepts |
|---|---|
| Basic operations $\pmod{n}$, inverses $\pmod{n}$ | Existence (when $\gcd(a, n) = 1$); computation (extended Euclidean algorithm) |
| Chinese Remainder Theorem (CRT) | Solving systems of congruences; applications in number theory and cryptography |
| Fermat / Euler / Wilson | Theorems; proofs; problem-solving applications |
| Polynomials mod $p$ | Roots, factorization; applications to number theory problems |
| **Residues / Primitive Roots** | |
| Primitive roots | Existence modulo primes; modulo $p^n$; computation |
| Quadratic residues | Properties; Legendre symbol; Euler's criterion |
| Quadratic reciprocity | Law of quadratic reciprocity; applications |
| Multiplicative order $\pmod{n}$ | Definition; computation; relation with primitive roots and cyclic groups |
| **Diophantine Equations** | |
| Factorization Methods | Difference of squares, Sophie Germain identity, special factorizations; Unique Factorization Domains (Gaussian, Eisenstein integers); Norms in algebraic number fields; Vieta jumping |
| Modular Arithmetic & Congruences | Reductions modulo primes or powers; Quadratic residues, Legendre symbol; Multiplicative order & primitive roots; Hensel lifting; Local–global principles (solvability mod $p$) |
| Parametrization of Solutions | Pythagorean triples; Rational parametrization of conics (general quadratics); Higher-degree parametrizations (elliptic curves, quartics) |
| Inequalities & Size Arguments | Bounding arguments; Infinite descent; Minimal solutions (no smaller solution possible) |
| Special Equations | Pell's equation: continued fractions, fundamental solution, recurrence; Fermat-type: $x^4 + y^4 = z^2$, |
| Descent & Structural Methods | Infinite descent; Descent on elliptic curves; Geometry of numbers |
| **Arithmetic Functions** | |
| Euler's totient's function | Properties, applications |
| Number / Sum of divisors | Computation, properties |
| Sum of digits | Basic properties |
| Möbius inversion | Definition, applications |
| **Algebraic Number Theory** | |
| Algebraic numbers | Minimal polynomials, field extensions, solving Diophantine equations |

## C  DATASET EXAMPLES AND ADDITIONAL STATISTICS

We present a few examples from `MathNet-Solve`, `MathNet-Retrieve`, and `MathNet-RAG` below. To explore the full dataset, we recommend visiting mathnet.mit.edu/explorer.

---

**Example from MathNet-Solve with figures**

**Problem.** Andrew and Olesya take turns cutting out either a $2 \times 2$ square or a $1 \times 1$ square from a $2 \times 2n$ rectangle, always along the grid lines, in such a way that after every move the remaining figure stays connected. The player who cannot make a move loses. Determine who wins if both play optimally and Olesya moves first.

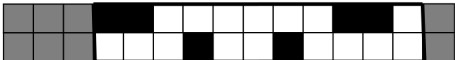

**Solution.** We proceed by induction on $n$.
For $n = 1$, the board is $2 \times 2$, and Olesya wins immediately by removing the whole $2 \times 2$ square. For $n = 2$, the board is $2 \times 4$, and Andrew wins by direct case analysis. Now suppose $n = 2m+1$ is odd. Then Olesya can remove a $2 \times 2$ square from one end of the strip, reducing the game to the case of even $n$. Thus it remains to prove that for even $n$, the second player has a winning strategy.
Consider therefore a board of size $2 \times 2k$. Until and including the first time the first player cuts a $1 \times 1$ square from a bordering column of the active strip, the second player follows a symmetric strategy. This is always possible, and the first such move must occur when the active playing field has size $2 \times \ell$ with $\ell$ even.
From that point onward, it suffices for the second player to prevent the first player from ever removing a $2 \times 2$ square. Indeed, once this is impossible, all remaining moves consist of removing $1 \times 1$ squares, so the game ends after an even number of further moves, which is favorable to the second player.
The only positions from which a $2 \times 2$ square can be removed are the two ends of the active strip. Hence the second player simply ensures that no end becomes a removable $2 \times 2$ block. Concretely, whenever there is a "free" end, he removes one $1 \times 1$ square from that bordering $2 \times 2$ block. There can never be more than one free end at a time: Olesya cannot create two in a single move, and initially there are none.
If there is no free end, then one of the following applies at a bordering column:

1. If the bordering column has no removed square, Andrew removes one unit square from it.

2. If the bordering column has exactly one removed square:

   a) if the second-to-border column also has a removed square, Andrew removes the second square from the bordering column;

   b) if the second-to-border column has no removed square, then:

      i. if the third-to-border column is empty, or has a removed square in the same row, Andrew removes a unit square from the second-to-border column in that row;

      ii. if the third-to-border column has a removed square in the other row, Andrew removes a unit square from the bordering column, shrinking the active strip without creating a free end.

In every case, Andrew can respond so that no removable $2 \times 2$ square appears at an end. Therefore the parity of the remaining $1 \times 1$ moves is controlled in his favor, and the second player wins for even $n$.
Combining this with the odd case, we conclude:

> Olesya wins when $n$ is odd, and Andrew wins when $n$ is even.

**Metadata.** Author: Bogdan Rublyov.

---

**Example from MathNet-Solve**

**Problem.** Given positive integers $m$ and $n$, find the smallest integer $N$ ($\geq m$) with the following property: if an $N$-element set of integers contains a complete residue system modulo $m$, then it has a nonempty subset whose sum is divisible by $n$.

**Answer.**

$$N = \max\left\{m,\ m+n-\frac{1}{2}m\big((m,n)+1\big)\right\}.$$

**Solution.** Let $d = (m,n)$, and write $m = dm_1$, $n = dn_1$.
We first prove the lower bound

$$N \geq \max\left\{m,\ m+n-\frac{1}{2}m(d+1)\right\}.$$

Assume $n > \frac{1}{2}m(d+1)$. Choose a complete residue system modulo $m$, say $x_1,\ldots,x_m$, whose residues modulo $n$ consist of exactly $m_1$ copies of each of $1,2,\ldots,d$. For instance, one may take

$$i + dn_1 j, \qquad i = 1,2,\ldots,d, \quad j = 1,2,\ldots,m_1.$$

Now let $k = n - \frac{1}{2}m(d+1) - 1$, and choose $k$ further integers $y_1,\ldots,y_k$, all congruent to $1$ modulo $n$. Then

$$A = \{x_1,\ldots,x_m,y_1,\ldots,y_k\}$$

contains a complete residue system modulo $m$. However, no nonempty subset of $A$ has sum divisible by $n$: indeed, the sum of the least nonnegative residues modulo $n$ of all elements of $A$ is at least $1$ and at most

$$m_1(1 + 2 + \cdots + d) + k = \frac{1}{2}m_1 d(d+1) + k = n - 1.$$

Hence no nonempty subset sum can be congruent to $0$ modulo $n$. Therefore

$$N \geq m + n - \frac{1}{2}m(d+1),$$

which gives the claimed lower bound.

Next we show that

$$N = \max\left\{m,\ m+n-\frac{1}{2}m(d+1)\right\}$$

does have the required property.
We need the following standard fact: among any $k$ integers, there exists a nonempty subset whose sum is divisible by $k$. Indeed, for integers $a_1,\ldots,a_k$, let

$$S_i = a_1 + \cdots + a_i \qquad (1 \leq i \leq k).$$

If some $S_i \equiv 0 \pmod{k}$, we are done. Otherwise, two of the $S_i$ have the same residue modulo $k$, say $S_i \equiv S_j \pmod{k}$ with $i < j$, and then

$$a_{i+1} + \cdots + a_j = S_j - S_i \equiv 0 \pmod{k}.$$

A useful corollary is: among any $k$ integers each divisible by $a$, one can find a nonempty subset whose sum is divisible by $ka$.
We now split into two cases.

*Case 1:* $n \leq \frac{1}{2}m(d+1)$, so $N = m$.
Let $x_1,\ldots,x_m$ be a complete residue system modulo $m$. Since $m = dm_1$, these may be partitioned into $m_1$ groups, each of which is a complete residue system modulo $d$. Call a finite set of integers a *$d$-set* if the sum of its elements is divisible by $d$.
If $d$ is odd, write a complete residue system modulo $d$ as $y_1,\ldots,y_d$ with $y_i \equiv i \pmod{d}$. Then each such group can be partitioned into $\frac{d+1}{2}$ $d$-sets, for example

$$\{y_1,y_{d-1}\},\ \{y_2,y_{d-2}\},\ \ldots,\ \left\{y_{\frac{d-1}{2}},y_{\frac{d+1}{2}}\right\},\ \{y_d\}.$$

(see next page) Thus altogether we obtain $\frac{1}{2}m_1(d+1)$ $d$-sets. Since

$$n_1 \leq \frac{1}{2}m_1(d+1),$$

the corollary implies that some collection of these $d$-sets has total sum divisible by $n_1 d = n$.

If $d$ is even, similarly each complete residue system modulo $d$ yields $\frac{d}{2}$ $d$-sets, with one element left over; pairing leftovers from two such groups gives one more $d$-set. In total, from $x_1, \ldots, x_m$ we obtain

$$\frac{1}{2}m_1 d + \left\lfloor \frac{m_1}{2} \right\rfloor$$

$d$-sets. Since

$$n_1 \leq \frac{1}{2}m_1(d+1) = \frac{1}{2}m_1 d + \frac{1}{2}m_1,$$

we again have enough $d$-sets to apply the corollary and obtain a nonempty subset whose sum is divisible by $n$.

*Case 2:* $n > \frac{1}{2}m(d+1)$, so

$$N = m + n - \frac{1}{2}m(d+1).$$

Let $A$ be an $N$-element set containing a complete residue system modulo $m$, say $x_1, \ldots, x_m$, together with $n - \frac{1}{2}m(d+1)$ additional elements. If $d$ is odd, then as in Case 1 the elements $x_1, \ldots, x_m$ can be partitioned into

$$\frac{1}{2}m_1(d+1)$$

$d$-sets. Partition the remaining elements arbitrarily into

$$n_1 - \frac{1}{2}m_1(d+1)$$

groups of size $d$. From each such group, by the corollary, one can extract a $d$-set. Hence altogether we obtain exactly $n_1$ $d$-sets. Applying the corollary once more to the sums of these $n_1$ $d$-sets, we find a nonempty union of them whose total sum is divisible by $n_1 d = n$.

If $d$ is even, then from $x_1, \ldots, x_m$ we obtain

$$\frac{1}{2}m_1 d + \left\lfloor \frac{m_1}{2} \right\rfloor$$

$d$-sets, as in Case 1. If $m_1$ is even, the remaining elements can be grouped so as to produce another

$$n_1 - \frac{1}{2}m_1(d+1)$$

$d$-sets, giving again a total of $n_1$ $d$-sets.

If $m_1$ is odd, one element $x_i$ remains with $d \mid x_i - \frac{d}{2}$. Partition the other remaining elements into $2n_1 - m_1(d+1)$ groups of size $\frac{d}{2}$. Each such group contains a subset whose sum is divisible by $\frac{d}{2}$, and any two such $\frac{d}{2}$-sets combine to form a $d$-set. Together with $\{x_i\}$, this again yields enough $d$-sets to obtain $n_1$ of them in total. Applying the corollary as above gives a nonempty subset with sum divisible by $n$.

Therefore, in all cases,

$$N = \max \left\{ m, \ m + n - \frac{1}{2}m\big((m,n)+1\big) \right\}.$$

**Metadata.** Country: China, Competition: China Math Olympiad, Year: 2013

**Example from MathNet-Solve**

**Problem.** Fatima and Asma are playing the following game. First, Fatima chooses 2013 pairwise different numbers, called $a_1, a_2, \ldots, a_{2013}$. Then, Asma tries to know the value of each number $a_1, a_2, \ldots, a_{2013}$. At each time, Asma chooses $1 \leq i < j \leq 2013$ and asks Fatima "What is the set $\{a_i, a_j\}$?" (For example, if Asma asks what is the set $\{a_1, a_2\}$, and $a_1 = 17$ and $a_2 = 13$, Fatima will answer $\{13, 17\}$). Find the least number of questions Asma needs to ask, to know the value of all the numbers $a_1, a_2, \ldots, a_{2013}$.

**Solution.** The least number of subsets is 28. Suppose that Bibi has 3 lists 1, 2, 3 which enable her to find $N$ with certainty. Let the lists contain $a_1, a_2, a_3$ subsets respectively. For list $i = 1, 2, 3$ Alex announces the number $x_i$ of subsets in the list that contain $N$, and the ordered triple $x_1, x_2, x_3$ is the only information Bibi obtains. So being able to guess $N$ with certainty means that each triple $x_1, x_2, x_3$ yields a certain $N \in \{1, 2, ..., 1001\}$ as a solution. Because there are 1001 possible numbers $N$ and Alex can choose any of them, it is then necessary that the number of different triples $x_1, x_2, x_3$ is at least 1001. This number equals $(a_1 + 1)(a_2 + 1)(a_3 + 1)$ as there are $a_i + 1$ possible values of $x_i$, namely $0, 1, ..., a_i$ ($i = 1, 2, 3$). Hence we must have

$$(a_1 + 1)(a_2 + 1)(a_3 + 1) \geq 1001.$$

Now use the AM-GM inequality to estimate the total number $a_1 + a_2 + a_3$ of subsets used by Bibi:

$$1001 \leq (a_1 + 1)(a_2 + 1)(a_3 + 1) \leq \left(\frac{a_1 + a_2 + a_3}{3} + 1\right)^3$$

It follows from here that $a_1 + a_2 + a_3 \geq 28$. Indeed if $a_1 + a_2 + a_3 \leq 27$ then the right-hand side of the displayed inequality is at most $\left(\frac{27}{3} + 1\right)^3 = 1000$.

Now we show that 3 lists with a total of 28 sets suffice. Use the factorization $1001 = 7 \cdot 11 \cdot 13$, consider a $7 \times 11 \times 13$ parallelepiped with the numbers $1, 2, ..., 1001$ written in its unit cubes. Let the vertical dimension be 13, then there are 13 horizontal layers of unit cubes (of dimensions $7 \times 11$). For $i = 1, 2, ..., 12$ let $S_i$ be the union of the first $i$ horizontal layers. Let list 1 consist of the 12 sets $S_1, S_2, ..., S_{12}$, and let Alex say that $x_1$ of them contain $N$. Observe that this answer enables Bibi to determine the horizontal layer containing $N$, whatever the announced value $x_1 \in \{0, 1, ..., 12\}$. Indeed, since $S_1 \subset S_2 \subset ... \subset S_{12}$, the answer $x_1 = 0$ means that $N$ is in layer 13, $x_1 = 1$ means that it is in layer 12, etc. In general $x_1 = i \in \{0, 1, ..., 12\}$ implies that $N$ is in horizontal layer $13 - i$. Thus a list of 12 sets is enough to single out the necessary layer out of 13 possible ones. Analogous lists in the other two directions, with $7 - 1 = 6$ sets and $11 - 1 = 10$ sets, can determine the layers in those directions that contain $N$.

As a result $N$ becomes known with $12 + 6 + 10 = 28$ sets, with 3 lists used. If Bibi uses 2 lists with $a_1$ and $a_2$ sets then by the same reasoning it is necessary that

$$1001 \leq (a_1 + 1)(a_2 + 1) \leq \left(\frac{a_1 + a_2}{2} + 1\right)^2, \text{ hence } a_1 + a_2 \geq 61.$$

Finally it is clear that one list alone would require at least 1000 sets.

**Metadata.** Country: Saudi Arabia. Competition: JBMO TST. Year: 2017. Source: SAUDI ARABIAN MATHEMATICAL COMPETITIONS 2017. Authors: Sultan Albarakati, Lê Anh Vinh, Tarek Salama, Lê Phúc L. Organization: Mawhiba.

---

**Example from MathNet-RAG**

**Target Problem** Let $P(x)$ be a non-zero polynomial with integer coefficients, prove that the set of primes dividing $x_n = 2^n + P(n)$ is infinite.

---

**Retrieved Problem (Expert-RAG).** The sequence of positive integers $\{a_n\}_{n \geq 1}$ is such that $a_n \leq 1000^{1000\sqrt{\log_2 n}}, \quad \forall n = 1, 2, 3, \ldots$ It is known that any number appears in the sequence at most 5000 times. Prove that there are infinitely many primes, each of which divides at least one member of the sequence.

**Solution.** Suppose, for contradiction, that only finitely many primes divide the sequence $\{a_n\}$. Let these primes be $p_1, p_2, \ldots, p_k$.
Then every $a_n$ is a product of powers of these primes, i.e., $a_n = p_1^{\alpha_1} p_2^{\alpha_2} \cdots p_k^{\alpha_k}$ for some non-negative integers $\alpha_i$.
The number of positive integers less than or equal to $M$ that are composed only of these $k$ primes is at most $C_k \log^k M$ for some constant $C_k$ (since the number of $k$-tuples $\alpha_i$ with $p_1^{\alpha_1} \cdots p_k^{\alpha_k} \leq M$ is $O(\log^k M)$).
For each such number, it can appear at most 5000 times in the sequence. Thus, for $N$ large, the total number of distinct $a_n$ among the first $N$ terms is at most $5000 \cdot C_k \log^k M$, where $M = 1000^{1000\sqrt{\log_2 N}}$.
But $\log M = 1000\sqrt{\log_2 N} \cdot \log 1000$, so $\log^k M = O((\sqrt{\log_2 N})^k) = O((\log_2 N)^{k/2})$.
Thus, the total number of possible $a_n$ among the first $N$ terms is $O((\log_2 N)^{k/2})$, but $N$ grows much faster than this. This contradicts the fact that the sequence $\{a_n\}$ has one term for each $n$. Therefore, there must be infinitely many primes dividing some $a_n$. $\square$

---

**Example from MathNet-RAG**

**Target Problem** Fix a positive integer $d$. Yael and Ziad play a game as follows, involving a monic polynomial of degree $2d$. With Yael going first, they take turns to choose a strictly positive real number as the value of one of the coecients of the polynomial. Once a coefficient is assigned a value, it cannot be chosen again later in the game. So the game lasts for $2d$ rounds, until Ziad assigns the final coefficient. Yael wins if $P(x) = 0$ for some real number $x$. Otherwise, Ziad wins. Decide who has the winning strategy.

---

**Retrieved Problem (Expert-RAG).** Grisho fills one of the $*$s in the equation with a positive real number and Misha fills one of them with a positive real number until all the 12 $*$s finished. Prove that Misha can play in a way that the resulting equation has a real root.

$$*x^{12} + \cdots + *x^7 + x^6 + *x^5 + \cdots + *x + * = 0.$$

**Solution.** If the equation has the form $a_{12}x^{12} + \cdots + a_0 = 0$. Partition the coefficients into pairs $(a_0, a_1), (a_2, a_3), \ldots, (a_{11}, a_{12})$. Note that for all $b > 0$; $\min\left(bx^{2k} + cx^{2k-1}\right) = \left(\frac{c}{2k}\right)^{2k} \left(\frac{1-2k}{b}\right)^{2k-1}$. We then prove the following lemma.
Lemma: If $b$ or $c$ is a fixed positive real number. We can choose a positive real number instead of the other one such that $x^6 + bx^{2k} + cx^{2k-1}$ has a real root.
Proof: Let $f(x) = x^6$ and $g(x) = -bx^{2k} - cx^{2k-1}$. Then, $g(x)$ would be positive in the interval $\left(-\frac{c}{b}, 0\right)$. So, We should only consider this interval. Let us assume that the maximum of $g(x)$ occurs at $x = r = -\frac{(2k-1)c}{2kb}$. Then, it suffices to have $r^6 + \left(\frac{c}{2k}\right)^{2k} \left(\frac{1-2k}{b}\right)^{2k-1} < 0$. That is, $\left(\frac{(2k-1)c}{2kb}\right)^6 + \left(\frac{c}{2k}\right)^{2k} \left(\frac{1-2k}{b}\right)^{2k-1} < 0$. Now, if $c$ is given and $k \geq 4$ then choose $b$ small enough. If $k = 1, 2, 3$, choose $b$ large enough. Further, if $b$ is given, again, if $k \geq 4$ then choose $c$ large enough and if $k = 1, 2, 3$ then choose $c$ small enough. This completes our proof.
So, after Grisha's first choice, Misha will fill the other coefficient of the corresponding pair such that $a_j r_0^j + a_{j+1} r_0^{j+1} + r_0^6 = 0$. For some $r_0 < 0$. After this, by the next moves of Grisha, Misha selects his coefficint such that $a_j + a_{j+1}r_0 = 0$. Then, $r_0$ is the root of the desired equation.

## PDFs Examples

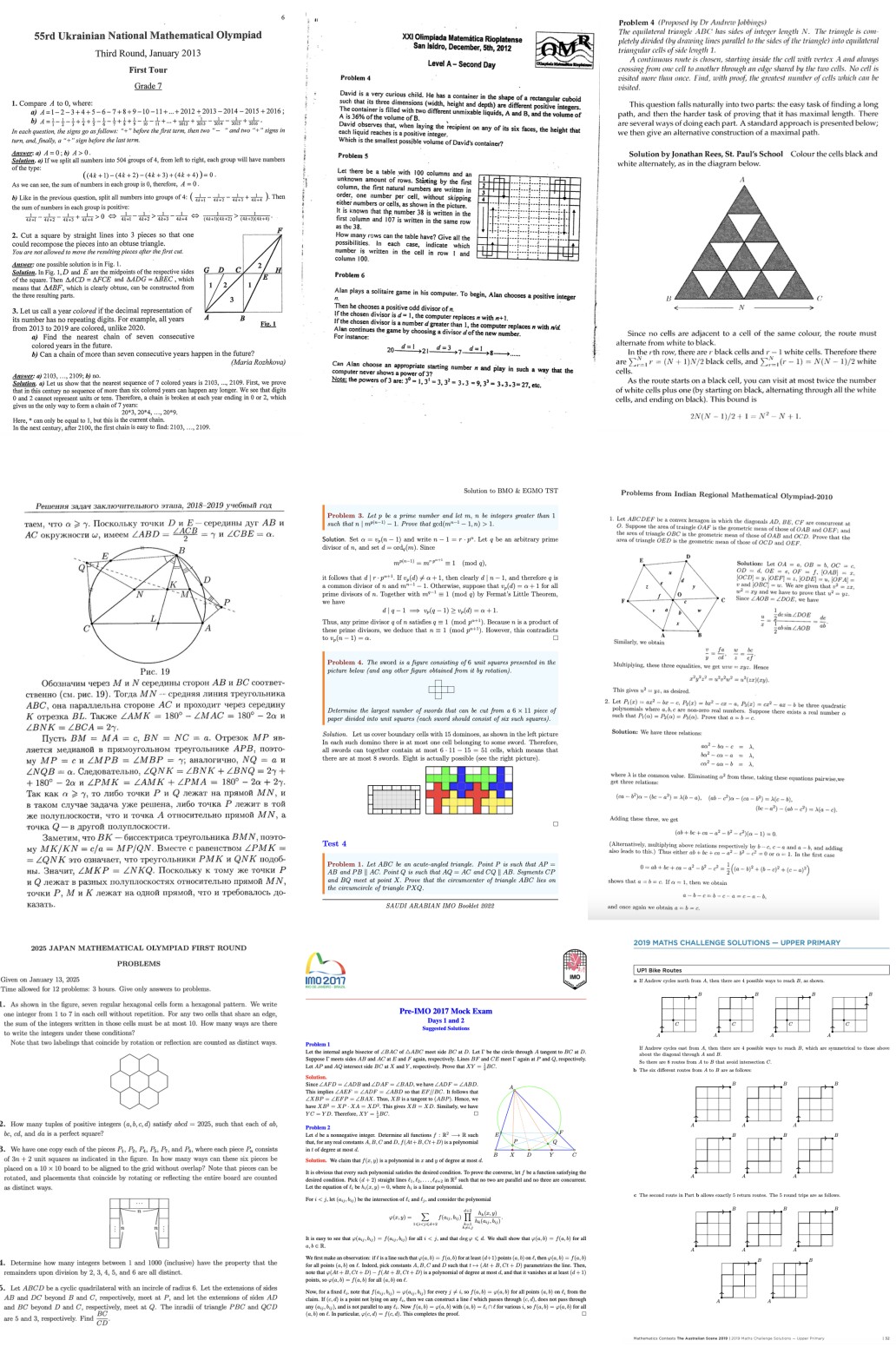

Figure 4: Examples of scanned pages from national mathematics Olympiad booklets across different countries and years. The examples shown are from Ukraine, Argentina, Russia, Saudi Arabia, India, Japan, and Australia.

Other Benchmarks Sources
*e.g. Omni-MATH* (ICLR 2025)

🟢 *MathNet* Sources
(official problems/solutions)

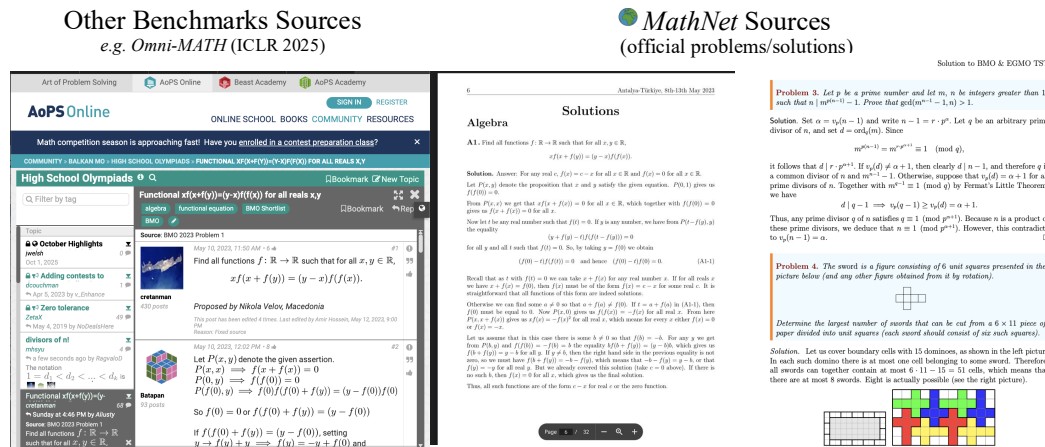

Figure 5: **MathNet is a collection of official Olympiad documents sourced directly from national problem booklets**. This example shows a BMO 2023 problem that appears in both MathNet and Omni-MATH Gao et al. (2024a) While Omni-MATH relies on the AoPS discussion shown on the left, MathNet provides the official problem and solution on the right.

## D    PERFORMANCE SENSITIVITY TO IMAGE PRESENCE, LANGUAGE

We study performance differences based on whether a problem contains figures and on the language of the sample. Table 9 reports results separately for samples with and without figures. Table 8 reports performance across different languages. Figure 6 shows cosine similarity distributions for equivalent problem pairs and hard negatives across different embedding models.

| *Problem Solving* Accuracy ($\%, \uparrow$) on `MathNet-Solve-Test` subsets | | | |
|---|---|---|---|
| **Model** | **Full Set** | **Text-only** | **Text+Images** |
| `gemini-3.1-pro-preview` | $\mathbf{78.4 \pm 1.0}$ | $\mathbf{76.7 \pm 1.2}$ | $\mathbf{85.1 \pm 1.9}$ |
| `gemini-3-flash-preview` | $\underline{70.4 \pm 1.1}$ | $\underline{67.8 \pm 1.3}$ | $\underline{80.4 \pm 2.1}$ |
| `gpt-5` | $69.3 \pm 1.1$ | $68.6 \pm 1.3$ | $72.0 \pm 2.4$ |
| `gpt-5-mini` | $57.0 \pm 1.2$ | $55.6 \pm 1.4$ | $62.5 \pm 2.6$ |
| `claude-opus-4-6` | $45.7 \pm 1.2$ | $43.8 \pm 1.4$ | $52.9 \pm 2.7$ |
| `gpt-5-nano` | $42.2 \pm 1.2$ | $45.1 \pm 1.4$ | $30.9 \pm 2.5$ |
| `gemini-2.5-flash` | $41.1 \pm 1.2$ | $38.0 \pm 1.3$ | $53.3 \pm 2.7$ |
| `DeepSeek-V3.2` | $40.1 \pm 1.2$ | $40.4 \pm 1.4$ | $38.7 \pm 2.7$ |
| `DeepSeek-R1` | $36.3 \pm 1.2$ | $35.4 \pm 1.3$ | $39.7 \pm 2.7$ |
| `grok-3` | $28.5 \pm 1.1$ | $27.0 \pm 1.2$ | $34.4 \pm 2.5$ |
| `gpt-4.1` | $21.4 \pm 1.0$ | $20.8 \pm 1.1$ | $23.6 \pm 2.2$ |
| `Llama-4-Maverick-17B-128E-Instruct-FP8` | $14.7 \pm 0.9$ | $14.1 \pm 1.0$ | $17.2 \pm 2.1$ |
| `gpt-4o` | $6.8 \pm 0.6$ | $6.0 \pm 0.6$ | $10.0 \pm 1.6$ |
| `Ministral-3B` | $4.4 \pm 0.5$ | $3.0 \pm 0.5$ | $10.0 \pm 1.6$ |

Table 8: **Modality-specific performance on `MathNet-Solve-Test`.** Results are reported as accuracy $\pm$ standard error on the full evaluation set (5,500 problems), the text-only subset (X problems), and the text+images subset (Y problems), with **best results** in bold and second-best results underlined. **Takeaway:** top multimodal reasoning models gain substantially on text+image problems.

| Model | Full Set | Chinese | English | French | Italian | Portuguese | Slovenian | Spanish |
|---|---|---|---|---|---|---|---|---|
| | | | *Problem Solving* Accuracy (%, ↑) on `MathNet-Solve-Test` | | | | | |
| `gemini-3.1-pro` | **78.4 ± 1.0** | **64.2 ± 8.1** | **77.4 ± 1.1** | **89.6 ± 8.3** | **96.6 ± 4.3** | **91.8 ± 2.7** | **84.0 ± 8.0** | **77.0 ± 10.7** |
| `gemini-3-flash` | 70.4 ± 1.1 | 43.1 ± 8.5 | 69.3 ± 1.2 | 72.9 ± 12.5 | **96.6 ± 4.3** | 89.3 ± 2.9 | 76.5 ± 9.3 | 72.1 ± 11.5 |
| `gpt-5` | 69.3 ± 1.1 | 45.5 ± 8.9 | 69.0 ± 1.2 | 60.4 ± 14.6 | 77.6 ± 11.2 | 82.5 ± 3.6 | 75.3 ± 9.3 | 65.6 ± 11.5 |
| `gpt-5-mini` | 57.0 ± 1.2 | 22.8 ± 7.3 | 56.5 ± 1.3 | 60.4 ± 13.5 | 75.9 ± 11.2 | 74.1 ± 4.1 | 63.0 ± 11.1 | 57.4 ± 13.1 |
| `claude-opus-4-6` | 45.7 ± 1.2 | 16.3 ± 6.5 | 44.7 ± 1.3 | 43.8 ± 14.6 | 62.1 ± 12.1 | 64.7 ± 4.7 | 55.6 ± 11.1 | 49.2 ± 13.1 |
| `gpt-5-nano` | 42.2 ± 1.2 | 15.4 ± 6.5 | 43.6 ± 1.3 | 45.8 ± 14.6 | 37.9 ± 12.1 | 36.7 ± 4.6 | 35.8 ± 10.5 | 36.1 ± 12.3 |
| `gemini-2.5-flash` | 41.1 ± 1.2 | 11.4 ± 5.3 | 39.6 ± 1.3 | 47.9 ± 14.6 | 62.1 ± 12.1 | 63.6 ± 4.6 | 67.9 ± 9.9 | 39.3 ± 12.3 |
| `DeepSeek-V3.2` | 40.1 ± 1.2 | 13.8 ± 6.1 | 39.9 ± 1.3 | 41.7 ± 14.6 | 53.4 ± 12.1 | 50.7 ± 4.7 | 44.4 ± 11.1 | 41.0 ± 12.3 |
| `DeepSeek-R1` | 36.3 ± 1.2 | 7.3 ± 4.5 | 35.8 ± 1.3 | 45.8 ± 14.6 | 50.0 ± 12.1 | 50.5 ± 4.7 | 39.5 ± 10.5 | 34.4 ± 11.5 |
| `grok-3` | 28.5 ± 1.1 | 4.1 ± 3.7 | 27.8 ± 1.2 | 18.8 ± 11.5 | 50.0 ± 12.9 | 43.5 ± 4.7 | 42.0 ± 11.1 | 29.5 ± 11.5 |
| `gpt-4.1` | 21.4 ± 1.0 | 2.4 ± 2.8 | 21.1 ± 1.1 | 16.7 ± 10.4 | 29.3 ± 12.1 | 31.1 ± 4.4 | 28.4 ± 9.9 | 19.7 ± 9.8 |
| `Llama-4-Maverick-17B` | 14.7 ± 0.9 | 3.3 ± 2.8 | 14.1 ± 0.9 | 8.3 ± 7.3 | 22.4 ± 10.3 | 26.6 ± 4.2 | 25.9 ± 9.9 | 9.8 ± 7.4 |
| `gpt-4o` | 6.8 ± 0.6 | 1.6 ± 2.0 | 6.2 ± 0.6 | 6.2 ± 7.3 | 6.9 ± 6.0 | 15.7 ± 3.4 | 19.8 ± 8.6 | 3.3 ± 4.1 |
| `Ministral-3B` | 4.4 ± 0.5 | 0.8 ± 1.2 | 3.8 ± 0.5 | 4.2 ± 5.2 | 3.4 ± 4.3 | 11.7 ± 3.0 | 21.0 ± 9.3 | 0.0 ± 0.0 |

Table 9: **Language-specific performance on `MathNet-Solve-Test`.** Results are reported as accuracy ± standard error across languages, with **best results** in bold and second-best results underlined. **Takeaway:** rankings are broadly consistent across languages, with `gemini-3.1-pro` and `gemini-3-flash` leading overall; performance is weakest on Chinese and strongest on Italian and Portuguese.

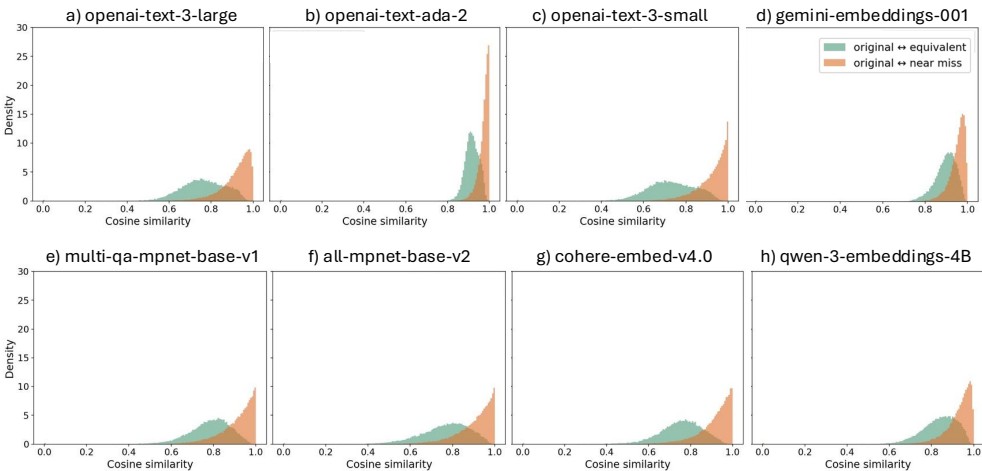

Figure 6: Cosine similarity distributions for equivalent (green) and near-miss/hard negatives (orange) problem pairs across different embedding models. Higher separation between the two distributions indicates a model's ability to distinguish structurally identical problems from those with small but critical alterations.

# E    LLM GRADERS VS HUMAN EXPERT GRADERS ON `MATHNET-RAG`

We benchmark the accuracy of a wide range of LLM graders and compare their judgments to human expert grading for *Retrieval-Augmented Problem Solving* on `MathNet-RAG`. This evaluation quantifies how reliably current models can act as automatic graders for Olympiad-level mathematical reasoning. For each model, we report *Retrieval-Augmented Problem Solving* performance under three settings: zero-shot, embed-RAG, and expert-RAG. This measures both cross-model grading consistency and alignment with human scoring.

| Accuracy (%, ↑) on `MathNet-RAG`: LLM vs. Human Expert Grading | | | | | | |
|---|---|---|---|---|---|---|
| **Model** | **LLM Grading** | | | | | **Human Grading** |
| | **LLaMA-4** | **DeepSeek-V3** | **GPT-4.1** | **GPT-4o** | **LLM Avg.** | **Human Expert** |
| **Zero-shot** | | | | | | |
| `claude-opus-4.5` | $72.2 \pm 7.6$ | $41.6 \pm 8.3$ | $31.4 \pm 7.8$ | $38.7 \pm 8.2$ | $46.0 \pm 8.4$ | $46.8 \pm 8.4$ |
| `deepseek-v3.2-speciale` | $96.2 \pm 3.2$ | $74.3 \pm 7.4$ | $85.5 \pm 6.0$ | $73.0 \pm 7.5$ | $82.2 \pm 6.5$ | $84.8 \pm 6.1$ |
| `gemini-3-pro-preview` | $94.7 \pm 3.8$ | $72.4 \pm 7.6$ | $71.7 \pm 7.6$ | $53.5 \pm 8.4$ | $73.1 \pm 7.5$ | $89.1 \pm 5.3$ |
| `gpt-5` | $98.1 \pm 2.3$ | $85.0 \pm 6.0$ | $83.2 \pm 6.3$ | $82.1 \pm 6.5$ | $87.1 \pm 5.7$ | $76.8 \pm 7.1$ |
| `grok-4.1-fast` | $92.7 \pm 4.4$ | $63.4 \pm 8.1$ | $76.5 \pm 7.2$ | $59.7 \pm 8.3$ | $73.1 \pm 7.5$ | $75.4 \pm 7.3$ |
| `olmo-3-32b-think` | $70.2 \pm 7.7$ | $44.3 \pm 8.4$ | $35.2 \pm 8.1$ | $48.2 \pm 8.4$ | $49.5 \pm 8.5$ | $45.2 \pm 8.4$ |
| `phi-4-reasoning-plus` | $46.2 \pm 8.4$ | $23.5 \pm 7.2$ | $6.6 \pm 4.2$ | $19.9 \pm 6.7$ | $24.1 \pm 7.2$ | $15.1 \pm 6.1$ |
| **Embed-RAG** | | | | | | |
| `claude-opus-4.5` | $64.9 \pm 8.1$ | $59.7 \pm 8.3$ | $40.3 \pm 8.3$ | $36.5 \pm 8.1$ | $50.3 \pm 8.5$ | $55.5 \pm 8.4$ |
| `deepseek-v3.2-speciale` | $94.2 \pm 3.9$ | $78.4 \pm 7.0$ | $92.2 \pm 4.5$ | $86.7 \pm 5.7$ | $87.9 \pm 5.5$ | $89.5 \pm 5.2$ |
| `gemini-3-pro-preview` | $95.9 \pm 3.3$ | $71.4 \pm 7.6$ | $68.6 \pm 7.8$ | $46.2 \pm 8.4$ | $70.5 \pm 7.7$ | $92.9 \pm 4.3$ |
| `gpt-5` | $93.2 \pm 4.2$ | $73.9 \pm 7.4$ | $81.0 \pm 6.6$ | $79.1 \pm 6.9$ | $81.8 \pm 6.5$ | $75.2 \pm 7.3$ |
| `grok-4.1-fast` | $88.6 \pm 5.4$ | $61.1 \pm 8.2$ | $72.2 \pm 7.6$ | $48.7 \pm 8.4$ | $67.7 \pm 7.9$ | $83.8 \pm 6.2$ |
| `olmo-3-32b-think` | $66.9 \pm 8.0$ | $38.6 \pm 8.2$ | $31.5 \pm 7.9$ | $45.3 \pm 8.4$ | $45.6 \pm 8.4$ | $54.6 \pm 8.4$ |
| `phi-4-reasoning-plus` | $36.6 \pm 8.1$ | $12.7 \pm 5.6$ | $8.2 \pm 4.6$ | $21.0 \pm 6.9$ | $19.6 \pm 6.7$ | $14.3 \pm 5.9$ |
| **Expert-RAG** | | | | | | |
| `claude-opus-4.5` | $77.1 \pm 7.1$ | $55.5 \pm 8.4$ | $53.6 \pm 8.4$ | $39.5 \pm 8.3$ | $56.4 \pm 8.4$ | $52.4 \pm 8.4$ |
| `deepseek-v3.2-speciale` | $97.1 \pm 2.8$ | $83.4 \pm 6.3$ | $89.3 \pm 5.2$ | $86.2 \pm 5.8$ | $89.0 \pm 5.3$ | $97.3 \pm 2.7$ |
| `gemini-3-pro-preview` | $99.6 \pm 1.0$ | $70.0 \pm 7.7$ | $72.7 \pm 7.5$ | $63.4 \pm 8.1$ | $76.4 \pm 7.2$ | $87.5 \pm 5.6$ |
| `gpt-5` | $97.3 \pm 2.7$ | $77.9 \pm 7.0$ | $82.6 \pm 6.4$ | $85.2 \pm 6.0$ | $85.8 \pm 5.9$ | $86.6 \pm 5.8$ |
| `grok-4.1-fast` | $92.5 \pm 4.5$ | $55.7 \pm 8.4$ | $74.2 \pm 7.4$ | $54.1 \pm 8.4$ | $69.1 \pm 7.8$ | $83.2 \pm 6.3$ |
| `olmo-3-32b-think` | $74.7 \pm 7.4$ | $49.0 \pm 8.4$ | $33.2 \pm 8.0$ | $47.4 \pm 8.4$ | $51.1 \pm 8.4$ | $47.6 \pm 8.4$ |
| `phi-4-reasoning-plus` | $48.6 \pm 8.4$ | $31.3 \pm 7.8$ | $9.7 \pm 5.0$ | $30.6 \pm 7.8$ | $30.0 \pm 7.7$ | $16.7 \pm 6.3$ |

Table 10: *LLM Grading* **vs. human expert grading on `MathNet-RAG` (35 problems).** For each solver and retrieval setting, we report the score assigned by four LLM graders, their mean (*LLM Avg.*), and the corresponding human expert score; all entries are accuracy (%) $\pm$ standard error. **Takeaway:** LLM graders broadly track the same solver ranking as human experts.

## F  LLM USAGE AND PROMPTS

We include the core prompts used for extraction, evaluation, and metadata classification. These are the exact versions used in our experiments.

Listing 1: System prompt for solution generation

```
SYSTEM_PROMPT = (
    "Solve the following math problem. Write out your full reasoning. "
    "At the very end, place your complete final response inside LaTeX \\
    boxed{}. "
    "- If the problem asks for a numerical or closed-form answer, put
    only that final expression in the box. "
    "- If the problem asks for a proof or argument, then enclose the
    entire proof (not just a concluding sentence) inside the box."
)
```

Listing 2: User prompt for grading

```
GRADING_PROMPT_TEMPLATE = """
You are an expert grader for the International Mathematics Olympiad (IMO).

Your task is to strictly and rigorously evaluate a proposed solution
    using official IMO standards.
Only fully justified, logically sound arguments may receive credit.

==================================================================
GENERAL SCORING RUBRIC (0-7)
==================================================================
```

```
- 7 points: Correct
A complete, correct, fully rigorous solution. Earlier incorrect attempts
    do not reduce the score if the final solution is fully correct.
-6 points: Almost Correct
The main idea is correct, and only minor gaps or small fixable errors
    remain. Missing major steps or relying on unjustified claims does not
     qualify.
- 1 point: Partial Progress
The student demonstrates substantial progress toward the solution.
Reformulations, trivial observations, or irrelevant partial results do
    not qualify.
- 0 points: Incorrect
The solution contains fundamental logical flaws or makes no significant
    progress. Fake proofs, circular arguments, or unjustified assumptions
     receive 0.

=====================================================================
PROBLEM TYPE: {problem_type}
=====================================================================

=====================================================================
PROBLEM STATEMENT
=====================================================================
{problem_statement}

=====================================================================
GROUND-TRUTH SOLUTION
=====================================================================
{solution}

=====================================================================
GROUND-TRUTH FINAL ANSWER
=====================================================================
{final_answer}

=====================================================================
STUDENT SOLUTION
=====================================================================
{student_answer}

=====================================================================
EVALUATION PROCEDURE
=====================================================================
1. Study the official solution to understand the essential steps and
    criteria for partial credit.
2. Analyze the student's solution line-by-line, identifying every gap,
    unjustified inference, incorrect claim, or flaw in logic.
3. Determine whether the student makes any nontrivial progress toward the
     solution.
4. Assign the score strictly according to the IMO rubric.

=====================================================================
CLARIFICATION: Meaning of 'final_answer_correct'
=====================================================================
'final_answer_correct' refers only to whether the student's final stated
    answer matches the ground-truth final answer (numerically,
    algebraically, or structurally).
The student's final answer is typically enclosed in \\boxed{{}} -- use
    that as the answer to compare.
It does NOT evaluate the correctness, completeness, or rigor of the
    student's reasoning.

Use 'doesn't apply' in either of these cases:
- the problem type is 'proof only' (no final answer expected), OR
- the student gives no \\boxed{{}} answer.
```

```
Examples:
- Correct \\boxed{{}} answer but no valid argument -> 'yes'
- Incorrect \\boxed{{}} answer but mostly correct reasoning -> 'no'
- No \\boxed{{}} answer -> 'doesn\'t apply'
- Proof-only problem -> 'doesn\'t apply'

======================================================================
OUTPUT FORMAT (MUST FOLLOW EXACTLY)
======================================================================
Output exactly one JSON object:

{{
  'score': {{
    'points': 0,
    'label': '0 out of 7'
  }},
  'analysis': {{
    'detailed_reasoning': '',
    'identified_errors': [],
    'partial_progress_assessment': ''
  }},
  'meta': {{
    'final_answer_correct': 'yes | no | doesn\'t apply',
    'contains_logic_errors': 'yes | no'
  }}
}}

RULES
- You must output nothing outside the JSON.
- 'points' must be one of: 0, 1, 6, 7.
- 'label' must match exactly: 'X out of 7'.
- All fields are required.
"""
```

Listing 3: System prompt for topics, final answer, and metadata extraction

```
SYSTEM_PROMPT = r"""
You are a rigorous math-olympiad content analyzer.

You will be given one 'problem package' containing:
- A problem statement
- One or more official solutions (labeled 'Solution 1', "Solution 2",
    ...)
- Optional final answers

Your tasks are:

======================================================================
1. TOPIC EXTRACTION
======================================================================
- Assign the problem its most specific topics from the taxonomy.
- Each topic path must be an array of strings from general -> specific.
- Include ALL paths relevant to the problem or solutions.
- Every topic path must be a verbatim copy of a path from the taxonomy.
- No paraphrasing, renaming, reordering, or combining nodes.
- Every topic must begin with "Topics".

======================================================================
2. MAIN IDEAS / TRICKS / TOOLS
======================================================================
- Produce a bullet list of the key structural insights or tools used.
- Examples:
  - Techniques used
  - Classical lemmas or theorems applied
```

```
  - Core inequality strategies
  - Key constructions or combinatorial ideas
- Do NOT retell the whole solution; extract the essential tools.

========================================================================
3. NATURAL-LANGUAGE PROBLEM DESCRIPTION
========================================================================
- Summarize the core task of the problem in normal English.
- NO mathematical symbols at all (no variables, no equations, no angle
    notation, etc.)
- A high-level, intuitive, short description.

========================================================================
4. PROBLEM TYPE CLASSIFICATION
========================================================================
Classify the problem into exactly one of the following:

- "proof only": no explicit final numeric/closed-form answer is required.
- "final answer only": problem only asks for a value/choice with no proof
      required.
- "proof and answer": requires both reasoning and a final value/statement.

- "MCQ": problem requires choosing from given options.

========================================================================
5. FINAL ANSWER EXTRACTION
========================================================================
- If the problem requires a final numeric/closed-form expression, value,
    or choice, extract it.
- If the problem's nature does NOT require a final answer (e.g., proof-
    only), output `null`.

Specific rules:
- If multiple solutions exist, the final answer must match the official
    answer section if present.
- Accept integers, expressions, ranges, choices, constructed forms, etc.
- For MCQ, return the *selected option* if identifiable; otherwise null.

========================================================================
TAXONOMY BLOCK
========================================================================
Use this taxonomy for the "topics" field.
Each topic path must follow the hierarchy strictly.

|-- Topics
| |-- Geometry
| | |-- Plane Geometry
| | | |-- Triangles
| | | | |-- Triangle centers: centroid, incenter,
| | | | |    circumcenter, orthocenter, Euler line,
| | | | |    nine-point circle
| | | | |-- Triangle inequalities
| | | | |-- Triangle trigonometry
| | | |-- Quadrilaterals
| | | | |-- Cyclic quadrilaterals
| | | | |-- Inscribed/circumscribed quadrilaterals
| | | | |-- Quadrilaterals with perpendicular diagonals
| | | |-- Circles
| | | | |-- Coaxal circles
| | | | |-- Tangents
| | | | |-- Radical axis theorem
| | | | |-- Circle of Apollonius
| | | |-- Concurrency and Collinearity
| | | | |-- Ceva's theorem
| | | | |-- Menelaus' theorem
```

```
        ... (more topics here)
======================================================================
OUTPUT FORMAT (STRICT JSON)
======================================================================

Return ONLY a JSON object:

{
  "topics": [
      ["Topics", "...", "..."],
      ["Topics", "...", "..."]
  ],
  "main_ideas": [
      "key idea 1",
      "key idea 2",
      "key idea 3"
  ],
  "natural_language_description": "...",
  "final_answer": "... or null",
  "problem_type": "proof only | final answer only | proof and answer |
    MCQ",
  "confidence": 0.0-1.0
}
Rules:
- NO text outside the JSON.
- NO markdown in the output.
- natural_language_description must contain zero mathematical symbols.
- Confidence reflects how certain you are about the classification.
"""
```

**LLMs Usage in the Paper** The authors made use of large language models (LLMs) primarily to support the writing process, including polishing the text for clarity and readability. In addition, LLMs were employed to assist in refining the design of the project website as well as the interface used by annotators.

