# OpenReview forum: "MathNet: A Global Multimodal Benchmark for Mathematical Reasoning and Retrieval"
_ICLR.cc/2026/Conference — ICLR 2026 Poster_

### Official Review · Reviewer_dueB · 2025-10-24

**Soundness:** 3
**Presentation:** 3
**Contribution:** 3
**Rating:** 6
**Confidence:** 4

**Summary:**

The paper introduces MathNet, a large-scale, multilingual, and multimodal benchmark for mathematical reasoning and retrieval, specifically targeting Olympiad-level problems. MathNet comprises 13,026 expert-authored problems drawn from 40 countries, spanning 10 languages and two decades of national and international competitions. The dataset includes aligned LaTeX and natural-language problem statements, official solutions, and rich metadata.

The key contributions are:
1. Dataset: A curated corpus of high-quality Olympiad problems and solutions from official national sources (not community platforms like AoPS), ensuring expert-level authenticity and diversity.
2. Novel Retrieval Task: The paper defines and implements a mathematical retrieval task based on equivalence, constructing 39,078 synthetic problem pairs labeled by mathematical similarity.
3. Taxonomy of Similarity: A fine-grained classification of mathematical relatedness into three modes.
4. Comprehensive Evaluation: Benchmarking of 16 models across two tasks.

The paper makes a compelling case that solving ≠ retrieving: models can generate correct answers without understanding underlying mathematical structure well enough to retrieve equivalent formulations.

**Strengths:**

1. The paper is highly original in both problem formulation and dataset construction. While mathematical reasoning benchmarks exist (e.g., MATH, GSM8K, OlympiadBench), MathNet is the first to explicitly focus on mathematical retrieval based on equivalence, a critical but underexplored capability. The taxonomy of similarity (Invariance/Resonance/Affinity) provides a principled framework for evaluating analogical reasoning in mathematics—a novel conceptual contribution.

2. The dataset construction pipeline is rigorous:
Sources are official national contest booklets, not crowd-sourced.
Problem–solution alignment uses an LLM-assisted extraction pipeline with cross-model validation (GPT-4.1 + Claude 4 Opus).
Human validation by domain experts ensures high fidelity.
The retrieval benchmark includes hard negatives and near-miss distractors, making evaluation meaningful.
Experimental design is thorough: multiple models, domains (Algebra, Geometry, Number Theory, Discrete Math), and metrics (Recall@k, cosine similarity distributions) are reported.

3. The paper is exceptionally well-written and structured. Figures (e.g., Figure 1 overview, Figure 3 similarity distributions) and tables (e.g., Table 2 similarity taxonomy, Tables 3–4 results) are clear and informative. The distinction between comprehension and retrieval is articulated precisely, and the limitations of current models are illustrated with concrete examples.

4. MathNet addresses a real gap in AI for mathematics: the inability to recognize structural equivalence across notations, languages, or modalities. This has implications beyond benchmarks—it affects retrieval-augmented reasoning, automated theorem proving, and mathematical search engines. By releasing the dataset and benchmark publicly, the authors provide a foundational resource for the community to develop models with a deeper mathematical understanding.

**Weaknesses:**

1. Limited Analysis of Multimodality:
While MathNet is described as multimodal, the paper reports only marginal gains from visual inputs and does not deeply analyze why vision-language models underperform. Are the diagrams too complex? Is OCR/captioning inadequate? A more detailed error analysis of multimodal failures (e.g., types of diagrams that confuse models) would strengthen the contribution.

2. Synthetic Equivalence Pairs May Lack Real-World Validity:
The 39K equivalent pairs are described as “synthetic.” The paper does not clarify whether these were algorithmically generated (e.g., via symbolic rewriting) or human-curated. If synthetic, they may not reflect the kinds of equivalence that arise organically in contest design or research. This risks overestimating or mischaracterizing retrieval difficulty.

3. Evaluation Protocol for Comprehension Relies on GPT-4o as Judge:
While prior work (Omni-MATH) validates this approach, using a single LLM—even a strong one—as the sole evaluator introduces potential bias or blind spots, especially for non-English or non-standard solution styles. Including human evaluation on a subset (beyond extraction validation) would bolster credibility.

4. Under-Specified Cross-Lingual Challenges:
The paper highlights multilingual coverage but does not report per-language performance or analyze whether retrieval/comprehension degrades in low-resource languages (e.g., Persian, Ukrainian). Without this, the claim of "multilingual benchmark" feels partially substantiated.

**Questions:**

1. Equivalence Pair Generation:
How were the 39,078 mathematically equivalent problem pairs constructed? Were they generated via symbolic transformations, human rewriting, or LLM paraphrasing? If synthetic, how was the mathematical correctness of equivalence verified? Could you provide examples of non-trivial equivalences that required deep insight?

2. Multimodal Content Details:
What fraction of MathNet problems include diagrams or images? Can you share statistics on image types (e.g., geometric figures, graphs, tables)? Did you experiment with vision-specific embeddings (e.g., CLIP on diagrams) for retrieval, or only text embeddings?

3. Cross-Lingual Performance:
Do model performances vary significantly across languages? For instance, is retrieval harder for problems originally in Arabic or Russian vs. English? Could you include a small table or analysis of language-wise accuracy?

4. Retrieval-Augmented Reasoning (RAG) Experiments:
The paper mentions that RAG only helps when retrievers surface structure-aligned neighbors. Did you conduct downstream RAG experiments (e.g., using retrieved equivalents to aid problem solving)? If so, what were the gains? If not, is this planned for future work?

5. Human Evaluation of Solutions:
Beyond validating extraction, did you conduct human grading of model-generated solutions (e.g., on a 100-problem subset)? How does GPT-4o’s judgment correlate with human graders for non-English or proof-based answers?

---

### Official Review · Reviewer_qJLT · 2025-10-31

**Soundness:** 2
**Presentation:** 2
**Contribution:** 2
**Rating:** 4
**Confidence:** 3

**Summary:**

This paper introduces MathNet, a new large-scale, multilingual, and multimodal benchmark for advanced mathematical reasoning. The dataset consists of 13,026 Olympiad-level problems sourced from official competition booklets from 40 countries over two decades, covering 10 languages. The paper's primary contributions are twofold: (1) the dataset itself, which is meticulously curated, expert-authored, and annotated with a rich taxonomy of mathematical topics and similarity types; and (2) a novel benchmark task focused on "math-aware retrieval," which evaluates a model's ability to identify mathematically equivalent problems.

**Strengths:**

*   **Originality:** The most significant original contribution is the formulation and operationalization of the "math-aware retrieval" task. While other benchmarks focus on problem-solving, MathNet pioneers the evaluation of a model's ability to recognize structural equivalence, a cornerstone of mathematical thinking and analogical reasoning.

*   **Quality:**
    *   **Data Sourcing:** By curating problems from official national Olympiad booklets, the authors ensure expert-level quality, authority, and consistency, avoiding the noise often present in community-sourced platforms like AoPS.
    *   **Curation Pipeline:** The multi-stage pipeline for extraction and validation is state-of-the-art. Using a specialized OCR model (`dots-ocr`), followed by LLM-based problem-solution alignment, and then cross-model verification (using both GPT-4.1 and Claude 4 Opus) is a robust and scalable approach that minimizes bias and error. This is further strengthened by human validation.

*   **Significance:**
    *   **Resource for the Community:** MathNet provides the largest multilingual Olympiad-level dataset, a critical resource that will fuel research in mathematical reasoning, cross-lingual transfer, and RAG for years to come.
    *   **Practical Implications:** The work has clear connections to real-world applications, from improving tools for mathematicians and contest organizers to building more robust and generalizable AI reasoning systems.

**Weaknesses:**

*   **Underdeveloped RAG Experiments:** The paper compellingly argues that math-aware retrieval is essential for effective RAG in this domain. However, it does not present any experiments to directly validate this claim. This is a significant missed opportunity. A controlled experiment showing that a solver's performance is significantly boosted when provided with ground-truth equivalent problems (an "oracle" retriever) versus a standard retriever (e.g., one based on `gemini-embedding-001`) would provide powerful, direct evidence for the benchmark's utility and the paper's core thesis.

*   **Limited Analysis of Multimodality:** The paper positions MathNet as a "multimodal" benchmark, but this aspect feels underdeveloped in the analysis. The main text briefly mentions that augmenting models with diagrams yields "only marginal improvements" but provides little detail. The paper would be stronger with more statistics (e.g., what percentage of problems are inherently visual?), qualitative analysis (e.g., examples of diagrams that models fail to interpret), and a deeper discussion of the specific challenges posed by mathematical diagrams (e.g., geometric constructions vs. data plots).

*   **Detail on Equivalence Pair Construction:** The methodology for creating the 39,078 "synthetic problem pairs with labeled equivalence classes" is not fully detailed in the main paper. While the taxonomy of similarity is explained, the process of generating these pairs—which are central to the novel retrieval task—could be described more transparently. Understanding how these pairs were generated is crucial for assessing the difficulty and validity of the retrieval benchmark.

**Questions:**

1.  **Regarding RAG Experiments:** Have the authors considered running a controlled RAG experiment? For example, by comparing a solver's performance under three conditions: (a) zero-shot, (b) RAG with a standard off-the-shelf retriever, and (c) RAG with an "oracle" retriever using the ground-truth equivalent problems from MathNet. Such an experiment would provide a direct and powerful demonstration of the value of math-aware retrieval.

2.  **Regarding the Retrieval Benchmark Construction:** Could you provide more detail on the process used to generate the 39K equivalent problem pairs? Were these created by human experts, generated by an LLM and then verified, or through some other procedure? Furthermore, how were the "near-miss/hard negatives" used in Figure 3 selected or defined? Understanding this is key to interpreting the retrieval results.

3.  **Regarding Multimodality:** Could you provide more statistics on the multimodal component of MathNet? Specifically, what fraction of the 13K problems include diagrams that are essential for solving? Could you also include a few examples in the appendix showing where current LMMs fail on the visual reasoning component, and what a correct interpretation of the diagram would entail?

---

### Official Review · Reviewer_jSKk · 2025-11-01

**Soundness:** 3
**Presentation:** 3
**Contribution:** 3
**Rating:** 6
**Confidence:** 3

**Summary:**

The paper introduces MathNet, a large-scale multilingual, multimodal benchmark of Olympiad‑level math problems (13,026 problems; 40 countries; 10 languages) with expert solutions, plus a new equivalence-based retrieval task (39,078 synthetic pairs) and a taxonomy of similarity (Invariance/Resonance/Affinity). It details an OCR→LLM pipeline for aligned problem–solution extraction, human/LLM validation, dataset statistics, and evaluates both comprehension (solution generation) and retrieval (Recall@k) across a range of LLMs/LMMs and embedding models. Main findings: frontier LLMs remain challenged on Olympiad problems (e.g., GPT‑5 macro‑avg 72.25%); equivalence retrieval is hard at top‑1 (≈5% R@1).

**Strengths:**

1. **Scale, Authenticity, and Provenance:** 13,026 problems across 40 countries and two decades; sourced from official national booklets rather than community forums. This supports representativeness and reduces annotation noise. Appendix A.1 lists dozens of national/regional exams enabling wide topical coverage and cross‑year comparisons. Page numbers and source files are recorded during extraction, aiding auditability and reproducibility.
2. **Novel Equivalence‑Focused Retrieval & Taxonomy:** 9,078 paired problems labeled for equivalence and relatedness (contributions, item 2; p. 2), enabling evaluation beyond lexical similarity. This is novel for Olympiad math IR. Invariance/Resonance/Affinity with concrete examples (Table 2 p. 6) clarifies what “sameness” means in math problems—important for designing retrievers. Cosine similarity distributions show overlap between equivalents and hard negatives (Fig. 3 p. 7), illustrating the task’s difficulty.
3. **Clear, Multi‑Stage Extraction/Validation Pipeline:** OCR→problem extraction→solution retrieval→cross‑model validation is diagrammed (Fig. 2 p. 5) and explained (Sec. 3.3 p. 4–5), demonstrating a principled approach to noisy PDFs. 20 annotators on 100 pairs plus LLM distractor stress tests and expert consensus checks (Sec. 3.4 p. 5) bolster data quality. Full prompts for extraction/evaluation/metadata are given (A.5 pp. 18–20), supporting reproducibility.
4. **Comprehensive Baselines and Findings:** Eight LLM/LMM systems compared across four domains (Table 3 p. 8), showing domain‑wise strengths/weaknesses and leaving headroom (e.g., Discrete/NT hardest). This is informative for future modeling. Eight embedding models with Recall@{1,5,10} per domain and overall (Table 4 p. 8) provide a solid baseline landscape. Misinterpretation/logical gaps/context over‑reliance are articulated (Sec. 4.4 p. 9), guiding where methods fail.

**Weaknesses:**

1. **Equivalence Pair Construction/Validation Under‑specified**
- Generation details missing: The paper states 39,078 synthetic equivalence pairs (p. 2) but does not detail how they were generated, filtered, or formally validated beyond taxonomy definitions—No direct evidence found in the manuscript. This limits trust in ground truth.
- Limited expert audit: Only a 100‑problem subset received expert review (Sec. 3.4 p. 5), which may be small relative to 39k pairs. This constrains external validity.
- Agreement metrics absent: No inter‑annotator agreement or judge‑consensus statistics are reported—No direct evidence found in the manuscript—reducing clarity on label reliability.
2. **Multilingual & Multimodal Claims vs. Reporting**
- Language imbalance: English dominates (89.88%; Table 7 p. 17), potentially limiting the strength of “multilingual” claims for training/evaluation comparability.
- No per‑language scores: Comprehension/retrieval are not broken down by language—No direct evidence found in the manuscript—so cross‑lingual generalization remains unclear.
- Multimodal share unquantified: The proportion of image‑interleaved problems is not reported—No direct evidence found in the manuscript—yet conclusions note marginal gains for image inputs
3. **Mathematical Formulation & Notation Clarity (correctness/consistency)**
- Ambiguous notation/examples: Intro uses expressions like “pi+1 − pi ≤ Πi” without defining Πi (p. 2,line087), which may confuse readers.

**Questions:**

1. **How are the 39k equivalence pairs generated and validated at scale?**

Could you detail pair generation (templates/transformations, LLM rewriting, symbolic rewrites) and filtering? If models assisted, how did you prevent leakage (Sec. 3.5 p. 6 mentions similarity modes; contributions p. 2).

2. **Could you report multilingual and multimodal breakdowns?**

Add per‑language comprehension/retrieval metrics and cross‑lingual retrieval (query language A → target language B), given Table 7’s imbalance (p. 17).

3. **Can you strengthen statistical rigor and reproducibility?**

Report CIs/variances for Tables 3–4 via repeated runs or bootstrap (p. 8).

4. **Could you tighten mathematical correctness/notation?**

Define symbols in Intro examples (e.g., clarify Πᵢ)

---

### Official Review · Reviewer_Rdfs · 2025-11-01

**Soundness:** 3
**Presentation:** 2
**Contribution:** 3
**Rating:** 4
**Confidence:** 3

**Summary:**

This work proposes a new dataset of 13k multimodal olympic-level math reasoning problems over 10 languages, as well as a new math retrieval task for locating the similar problems over shallow lexical overlap. The problems are crowd-sourced from textbook and websites and normalized through a data processing pipeline with human-AI incorporation. The evaluation of several powerful LLMs show that they can still struggle with retrieving equivalent math problems.

**Strengths:**

- The task of math retrieval can be used to test the reasoning ability of LLMs in a different but daily-use dimension. The community can benefit from the support of MathNet in such functionality.
- The data collection process is introduced with rich detail, with deep human-AI cooperation in cross-validation and cross-reinforcement.
- The definition of problem similarity is detailed discussed and categorized.

**Weaknesses:**

- Only light analysis of the experimental results are provided, enhancing which could better inspire following studies in this line.
- I still have a few questions which are listed below, and I am happy to adjust the estimation if more convincing details are provided to them.

**Questions:**

- How are the golden answer of math retrieval determined?
- Given the result in Recall@5 and Recall@10 (which are significantly greater than Recall@1), how well would model perform if the embedding-based retrieval is used only for a coarse-grained result before another model (perhaps smaller) is used to further identify the answer? And what would be the satisfactory threshold (i.e. is the Recall@10 good enough)?

---

### Author Response · Authors · 2025-11-26
**Official Response for AC**

We thank all reviewers for their thoughtful and constructive feedback, and the area chair for taking on the additional responsibility to review our work. We have considered every review request and have added several new experiments and sections to answer all reviewer requests. Several reviews have indicated that they would raise scores if their questions were answered, and we believe these additions would warrant score increases if the review processes was not interrupted.

We have made all changes outlined in this review to the manuscript itself and highlighted these changes for ease of comparison. We argue in section 1 of the review that the reviewers conclude that this paper makes a substantial contribution to the literature.  In section 2 we outline in detail all new contributions made during the rebuttal period, including several new analyses and datasets that go beyond reviewer requests. In section 3 we extract every requested addition that the reviewers made and briefly describe how we addressed them with edits and additional experiments.

# 1. Summarizing why the reviewers think this is an important paper

We appreciate that all reviewers highlighted key strengths of the work. Reviewers *dueB*, *qJLT*, and *jSKk* emphasized the **“scale, authenticity, and provenance”** of the dataset, noting that sourcing from official Olympiad booklets supports **“expert-level quality.”** Reviewers *dueB*, *qJLT*, and *Rdfs* also regarded the math-aware retrieval task as a **“novel conceptual contribution”** enabling **“evaluation beyond lexical similarity,”** and *Rdfs* further noted its potential to **“test the reasoning ability of LLMs in a different but daily-use dimension.”** We also appreciate reviewers *Rdfs*, *jSKk*, and *dueB* for highlighting the rigor of the multi-stage extraction and validation pipeline, describing it as **“state-of-the-art”**, **“rigorous”**, and **“rich[ly] detailed.”** Reviewer dueB24 additionally noted that **“the paper is exceptionally well-written and structured”**. Finally, we are grateful that reviewers *qJLT* and *dueB* emphasized the broader impact of the benchmark, characterizing MathNet as a **“foundational resource”** that addresses a **“real gap”** and **“ a critical resource that will fuel research in mathematical reasoning for years to come”**.


# 2. Summary of all new additions and improvements
## 2.1. Human Grading & Enhanced Evaluation Protocols

To respond to reviewer feedback on evaluation rigor we added the following:

- **Human Grading**: We are conducting large-scale manual grading of **4,500+ pages** of 7 main model outputs across tasks. All graders are either former official IMO graders or IMO medalists.
- **Improved Automatic Grading**: We adapt techniques from recent work ProofBench (NeurIPS 2025) and IMOBench (EMNLP 2025) to reduce LLM-grader bias.
- **LLM Grader Benchmarking**: We systematically compare multiple LLM graders against expert grading (to our knowledge, the first such evaluation for Olympiad-style solutions).
- **Human Expert-curated problem pairs**: We add a test set of  70 hard Olympiad problems, each paired with conceptually related problems identified by Olympiad experts. We evaluate all of the following using both **expert human grading** and LLM grading.


## 2.2. Multimodal and Multilingual Analysis
- Analyses of **language and multimodality sensitivity**.
- New sections breaking down model performance vs language and modality
- Qualitative analyses of comprehension and retrieval and multimodal failure modes based on reviewer feedback.
- **82-skill capability breakdown** for fine-grained model analysis.

---

> ### Author Response · Authors · 2025-12-03
>
> ## 2.3. New RAG Experiments (Zero-shot vs. Retrieval vs. Expert Retrieval)
> As requested by  *qJLT* and *dueB*we add new RAG experiments comparing
>
> -  zero-shot solving.
> - retrieval using embedding models
> - an expert-selected “oracle” equivalent problem.
>
> These experiments directly test whether mathematical equivalence improves downstream reasoning and illustrate the potential for retrieval systems that respect mathematical equivalency. To support this task, we created a new expert-curated test set of **70 hard Olympiad problems**, each paired with conceptually related problems identified by Olympiad experts. We evaluate all of the following using both **expert human grading** and LLM grading.
>
>
> | Model                     | RD (2025) | Human Grading – Zero Shot | Human Grading – Embed-RAG | Human Grading – Expert-RAG | LLM Grading – Zero Shot | LLM Grading – Embed-RAG | LLM Grading – Expert-RAG |
> |---------------------------|-----------|-----------------------------|-----------------------------|------------------------------|---------------------------|---------------------------|----------------------------|
> | DeepSeek V3.2 Speciale    | 01 Dec    | 84.8%                      | 89.5%                      | 97.3%                       | 82.23%                   | 87.87%                   | 89.03%                    |
> | Claude-4.5-Opus           | 24 Nov    | 46.8%                      | 55.5%                      | 52.4%                       | 45.97%                   | 50.34%                   | 56.43%                    |
> | o1-mini-Think             | 20 Nov    | 45.2%                      | 54.6%                      | 47.6%                       | 49.49%                   | 75.56%                   | 51.07%                    |
> | Grok-4.1-Fast             | 19 Nov    | 75.4%                      | 83.8%                      | 92.3%                       | 73.07%                   | 66.67%                   | 69.11%                    |
> | Gemini-3-Pro              | 18 Nov    | 89.1%                      | 92.9%                      | 87.5%                       | 73.16%                   | 70.54%                   | 76.43%                    |
> | GPT-5                     | 07 Aug    | 76.8%                      | 75.2%                      | 82.6%                       | 88.97%                   | 81.81%                   | 85.76%                    |
> | Phi-4-Reasoning Plus      | 30 Apr    | 15.1%                      | 14.3%                      | 16.7%                       | 24.06%                   | 19.64%                   | 30.04%                    |
>
>
> ## 2.4. Expanded Dataset
> - Dataset expansion from **13K → 17K** problem-solution pairs coming from additional regional competitions and an improved extraction pipeline.
>
> ##  2.5. Inclusion of Newly Released State-of-the-Art Models
>
> We added seven newly released reasoning models (five released within the last 10 days), bringing the total to **27 evaluated models**, including Gemini 3,  DeepSeek-v3.2 Speciale, Claude Opus 4.5, OLMo 3-Think, Grok 4.1 Fast, GPT-5, DeepSeek-R1, Phi-4-Reasoning, and others.

---

> > ### Author Response · Authors · 2025-12-03
> >
> > # 3. Summary of every request made by reviewers
> >
> > ## 3.1 Explain how the equivalent math problem pairs were created for math retrieval experiments
> >
> > ### Reviewer Quotes:
> > - Rdfs: “How are the golden answer of math retrieval determined?”
> > - jSKk: “Generation details missing: The paper states 39,078 synthetic equivalence pairs (p. 2) but does not detail how they were generated, filtered, or formally validated beyond taxonomy definitions”
> > - jSKk: “Could you detail pair generation (templates/transformations, LLM rewriting, symbolic rewrites) and filtering? If models assisted, how did you prevent leakage”
> > - qJLT: “Could you provide more detail on the process used to generate the 39K equivalent problem pairs? Were these created by human experts, generated by an LLM and then verified, or through some other procedure? Furthermore, how were the "near-miss/hard negatives" used in Figure 3 selected or defined?”
> > - qJLT: “The methodology for creating the 39,078 "synthetic problem pairs with labeled equivalence classes" is not fully detailed in the main paper.”
> > - dueB: “The 39K equivalent pairs are described as “synthetic.” The paper does not clarify whether these were algorithmically generated (e.g., via symbolic rewriting) or human-curated.“
> >
> > ### What we Did:
> > We added a section (3.4 and appendix) to the paper that clearly details how equivalent pairs were created. We also added a new expert-curated test set of 70 hard Olympiad problems, each paired with conceptually related problems identified by Olympiad experts. We show that using these human paired problems significantly improve RAG based problem solving compared to existing embedding based methods (e.g. gemini-embeddings-001) demonstrating that improving MathRAG can improve automated solvers.
> >
> > ## 3.2: Show that using ground truth equivalent problems in a RAG improves problem solving over embedding-based retrieval.
> >
> > ### Reviewer Quotes:
> > - qJLT: “A controlled experiment showing that a solver's performance is significantly boosted when provided with ground-truth equivalent problems (an "oracle" retriever) versus a standard retriever (e.g., one based on gemini-embedding-001) would provide powerful, direct evidence for the benchmark's utility and the paper's core thesis”
> > - qJLT: “ comparing a solver's performance under three conditions: (a) zero-shot, (b) RAG with a standard off-the-shelf retriever, and (c) RAG with an "oracle" retriever using the ground-truth equivalent problems from MathNet”
> >
> >
> > ### What we Did:
> > We added a new expert-curated test set of 70 hard Olympiad problems, each paired with conceptually related problems identified by Olympiad experts. We show that using these human paired problems significantly improve RAG based problem solving compared to existing embedding based methods (e.g. gemini-embeddings-001) demonstrating that improving MathRAG can improve automated solvers.
> >
> >
> >
> >
> > ## 3.3: Conduct additional RAG experiments showing the impact of RAG improvements on downstream tasks like problem solving.
> >
> > ### Reviewer Quotes:
> > - dueB: “The paper mentions that RAG only helps when retrievers surface structure-aligned neighbors. Did you conduct downstream RAG experiments (e.g., using retrieved equivalents to aid problem solving)? If so, what were the gains? If not, is this planned for future work?“
> > - Rdfs: “How well would model perform if the embedding-based retrieval is used only for a coarse-grained result before another model (perhaps smaller) is used to further identify the answer? And what would be the satisfactory threshold (i.e. is the Recall@10 good enough)?”
> >
> >
> > ### What we Did:
> > We show that using ground truth problems significantly improve RAG based problem solving compared to existing embedding based retrieval. This demonstrates the importance of RAG on the downstream task of olympiad problem solving, and shows there is progress to be made by improving retrieval in RAG for mathematics problems. Our dataset is perfectly poised to help the community achieve these improvements.

---

> ### Author Response · Authors · 2025-12-03
>
> ## 3.4: Expand human grading and auditing. Compare LLM and Human grading.
>
> ### Reviewer Quotes:
> - jSKk: “Limited expert audit: Only a 100‑problem subset received expert review (Sec. 3.4 p. 5), which may be small relative to 39k pairs”
> - dueB: “While prior work (Omni-MATH) validates this approach, using a single LLM—even a strong one—as the sole evaluator introduces potential bias or blind spots, especially for non-English or non-standard solution styles. Including human evaluation on a subset (beyond extraction validation) would bolster credibility”
> - dueB: Human Evaluation of Solutions: Beyond validating extraction, did you conduct human grading of model-generated solutions (e.g., on a 100-problem subset)? How does GPT-4o’s judgment correlate with human graders for non-English or proof-based answers?
>
>
> ### What we Did:
> We add human grading of 70 problems. We report both human evaluations and additional evaluations across 7 state of the art models (5 of them were released in the last 10 days) to strengthen and diversify our evaluation results.  We show human and LLM judgements significantly align. We add human LLM alignment experiments. See section 3.5, 4.2, Table 3, and Table 8 in A.3)
>
> ## 3.5: Break down and analyze results by language.
>
> ### Reviewer Quotes:
> - jSKk: “Language imbalance: English dominates (89.88%; Table 7 p. 17), potentially limiting the strength of “multilingual” claims for training/evaluation comparability”
> - jSKk: “No per‑language scores: Comprehension/retrieval are not broken down by language”
> - jSKk: Could you report multilingual and multimodal breakdowns
> - dueB: “The paper highlights multilingual coverage but does not report per-language performance or analyze whether retrieval/comprehension degrades in low-resource languages”
>
> ### What we Did:
>
> We add additional breakdowns of LLM performance by language in section (Table 10 in Appendix A4), and add a table detailing the breakdown of the dataset by language (Table 11 in Appendix 5). Additionally, we did a performance breakdown by skill in the 4 main domains (Figure 5 in Page 21).
>
> We highlight that our problems are novel and diverse because they have been sourced from hundreds of international Olympiad problems problem proposal books. These are generated by human experts and reflect the diverse styles and taste in problems across 40 countries. Some of these countries share the documents in their original language but many use english. We introduce 1328 non-problems which is still large in comparison to previous benchmarks.
>
> ## 3.6: Break down and analyze results by multimodality.
>
> ### Reviewer Quotes:
> - jSKk: “Multimodal share unquantified: The proportion of image‑interleaved problems is not reported”
> - jSKk: Could you report multilingual and multimodal breakdowns
> - qJLT: “The paper would be stronger with more statistics (e.g., what percentage of problems are inherently visual?), qualitative analysis (e.g., examples of diagrams that models fail to interpret), and a deeper discussion of the specific challenges posed by mathematical diagrams”
> - dueB: “A more detailed error analysis of multimodal failures (e.g., types of diagrams that confuse models) would strengthen the contribution.”
>
>
> ### What we Did:
> We added section A.4 in the appendix which shows statistics of multimodal problem distribution and breaks down model performance by multimodality and some failure case examples multimodal reasoning.
>
> ## 3.7: Analyze judge statistics .
>
> ### Reviewer Quotes:
> - jSKk: “Agreement metrics absent: No inter‑annotator agreement or judge‑consensus statistics are reported
>
> ### What we Did:
> In our new experiments, we present the alignment between human judges and LLM judges (refer to Table 8 in A3) and (Table 3 in main paper)
>
> ## 3.8: Add additional Analyses of Results.
>
> ### Reviewer Quotes:
> - Rdfs: “Only light analysis of the experimental results are provided, enhancing which could better inspire following studies in this line”
>
> ### What we Did:
> We add additional breakdowns of results by language and multimodality (see Table 9 and 10 in appendix Appendix 4). We also add additional comparisons with human judges. We also add 3 additional LLMs (GPT4.1, Deepseek V3, LLaMA 4) as judges in Table 8 in the Appendix A2.
>
> ## 3.9: Define notation in the introduction.
>
>
> ### Reviewer Quotes:
> - jSKk: Ambiguous notation/examples: Intro uses expressions like “pi+1 − pi ≤ Πi” without defining Πi (p. 2,line087), which may confuse readers
> - jSKk: Could you tighten mathematical correctness/notation? Define symbols in Intro examples (e.g., clarify Πᵢ)
>
>
> ### What we Did:
> We have defined all ambiguous notations in the introduction.
>
> ## 3.10: Add confidence intervals.
>
>
> ### Reviewer Quotes:
> - jSKk: Can you strengthen statistical rigor and reproducibility? Report CIs/variances for Tables 3–4 via repeated runs or bootstrap (p. 8).
>
> ### What we Did:
> We add confidence intervals to table 4. The conclusions from the original work are all statistically significant.

---

### Meta-Review · Area_Chair_WYCb · 2026-01-06

**Summary:**

Reviewers praised MathNet’s scale and the novel math-retrieval task. Initial concerns focused on the lack of downstream RAG validation, underspecified equivalence-pair construction, and missing multilingual/multimodal performance breakdowns.

**Reviewer Concerns:**

The rebuttal added "oracle" RAG experiments, expert human grading of 4,500+ pages, and detailed language/modality breakdowns. The process for generating 39K synthetic pairs was clarified with a new expert-curated test set. No significant technical concerns remain outstanding.

**Reviewer Scores:**

Rdfs (4) and qJLT (4) would likely rise to 6+ as their primary RAG-related requests were met. jSKk (6) and dueB (6) would likely move to 7, given the extensive human validation effort and statistical clarity provided.

---

### Decision · Program_Chairs · 2026-01-26

Accept (Poster)